# VGGT-SLAM: Dense RGB SLAM
# Optimized on the SL(4) Manifold

**Dominic Maggio*   Hyungtae Lim*   Luca Carlone**
Massachusetts Institute of Technology
{drmaggio, shapelim, lcarlone}@mit.edu

## Abstract

We present VGGT-SLAM, a dense RGB SLAM system constructed by incrementally and globally aligning submaps created from the feed-forward scene reconstruction approach VGGT using only uncalibrated monocular cameras. While related works align submaps using similarity transforms (*i.e.,* translation, rotation, and scale), we show that such approaches are inadequate in the case of uncalibrated cameras. In particular, we revisit the idea of reconstruction ambiguity, where given a set of uncalibrated cameras with no assumption on the camera motion or scene structure, the scene can only be reconstructed up to a 15-degrees-of-freedom projective transformation of the true geometry. This inspires us to recover a consistent scene reconstruction across submaps by optimizing over the $SL(4)$ manifold, thus estimating 15-degrees-of-freedom homography transforms between sequential submaps while accounting for potential loop closure constraints. As verified by extensive experiments, we demonstrate that VGGT-SLAM achieves improved map quality using long video sequences that are infeasible for VGGT due to its high GPU requirements. Our code is available at https://github.com/MIT-SPARK/VGGT-SLAM.

## 1   Introduction

One of the most fundamental tasks in computer vision is that of simultaneous localization and mapping (SLAM) where given multiple monocular (or stereo) images, the task is to generate a 3D reconstruction of the scene and estimate the 6-degrees-of-freedom (DOF) pose of the cameras. Most approaches for this have traditionally leveraged classical multi-view geometry constraints [47, 25, 80, 81], data association [38, 37], and backend optimization such as bundle adjustment [52, 53, 57, 1, 58, 59]. Recently, a new paradigm of using simpler, feed-forward networks, which produce point clouds from uncalibrated input images, has gained increasing popularity. In this thrust, the seminal work DUSt3R [73] takes in a pair of images and estimates dense point clouds of both images in the reference frame of the first camera, thus creating a dense scene reconstruction and allowing the camera poses estimated easily with a 3-point RANSAC solver [48, 40].

To extend feed-forward reconstruction to multiple images, VGGT (Visual Geometry Grounded Transformer) [71] takes in an arbitrary number of images, and in addition to estimating dense point clouds of each camera frame, also estimates depth maps, feature tracks, and camera poses and intrinsics. However, VGGT is limited in the number of images that can be processed by GPU memory. For example, in the case of an NVIDIA GeForce RTX 4090 with 24 GB, this is limited to approximately 60 images, making larger reconstructions requiring hundreds or thousands of images infeasible.

---

*Equal contribution.

39th Conference on Neural Information Processing Systems (NeurIPS 2025).

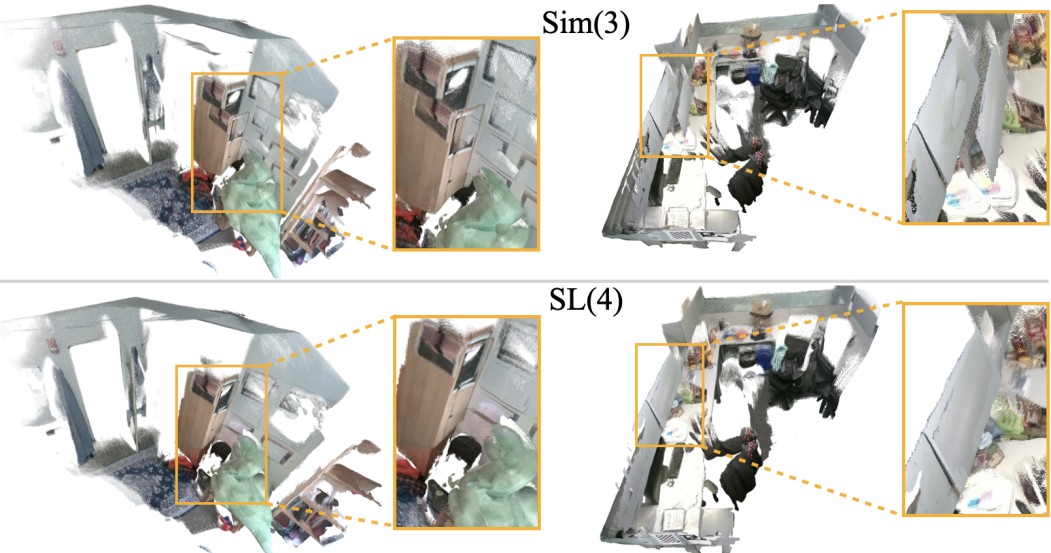

Figure 1: VGGT-SLAM alignment of 6 submaps created from VGGT using $\mathrm{Sim}(3)$ alignment (top) and $\mathrm{SL}(4)$ alignment (bottom). Here, $\mathrm{Sim}(3)$ is insufficient to align submaps due to a projective ambiguity, motivating our $\mathrm{SL}(4)$-based SLAM. Experiments performed on a segment of the Clio [39] apartment and cubicle scenes.

One may suspect that a simple, trivial solution would be to create multiple submaps with VGGT where each submap contains at least one overlapping image, and solve for the scale parameter between submaps (as the reconstruction does not capture metric scale), with VGGT's estimated poses being used to align rotation and translation (*i.e.,* estimating a $\mathrm{Sim}(3)$ transformation between submaps). While we demonstrate $\mathrm{Sim}(3)$ optimization shows impressive reconstructions in many cases, we empirically observe that the feed-forward nature of VGGT with uncalibrated cameras introduces a *projective ambiguity*, which in addition to the $\mathrm{Sim}(3)$ DOF includes shear, stretch, and perspective DOF, especially when the disparity between images becomes small. This ambiguity cannot be fully resolved through a similarity transformation alone.

Unveiling why a similarity transformation is sometimes insufficient for this recent transformer-based scene reconstruction method causes us to return to classical computer vision for answers, specifically the notion of projective ambiguity. To rectify a projective reconstruction to a metric reconstruction requires computing a $4 \times 4$ homography matrix [26] which can be mapped to the Special Linear, $\mathrm{SL}(4)$, Lie group. Since this is a Lie group, we can formulate the submap alignment problem as a factor graph optimized on the $\mathrm{SL}(4)$ manifold to globally align an arbitrary number of submaps given both estimates of relative homographies between sequential submaps and added constraints from detected loop closures.

**Contributions** Firstly, we present the first SLAM system that leverages the feed-forward scene reconstruction capabilities of VGGT [71], extending it to large-scale scenes that cannot be reconstructed from a single inference of VGGT. Our system operates entirely with monocular RGB cameras and does not require known camera intrinsics or consistent calibration across frames. Importantly, it achieves this without any additional training.

Secondly, while $\mathrm{Sim}(3)$ optimization is often sufficient, we identify and analyze scenarios where projective ambiguity arises, as presented in Fig. 1. In these cases, conventional similarity transforms do not fully resolve scale and alignment issues. We highlight this limitation and demonstrate how incorporating projective constraints addresses the problem.

Finally, we propose the first factor graph formulation that operates directly on the $\mathrm{SL}(4)$ manifold to address projective ambiguity. Even in practical scenarios, where projective ambiguity is less dominant, we show that $\mathrm{SL}(4)$-based optimization achieves performance competitive with or superior to other state-of-the-art learning-based SLAM approaches, offering a principled framework for handling cases where similarity transformations are insufficient.

## 2 Related Work

**Classical Scene Reconstruction**    Classical scene reconstruction methods typically rely on geometric features to estimate camera poses and reconstruct 3D scenes from multi-view images [45, 5, 19, 11, 56] using [32, 75] bundle adjustment [58, 50], by performing sparse feature extraction, matching, and robust pose estimation by optimizing for $SE(3)$ transformations. Several works have also performed dense SLAM [30, 76, 41, 77], and [6] provides a survey on SLAM. Multiple works have also performed classical projective scene reconstruction using uncalibrated cameras [68, 27]. Recently, Sim-Sync [78] introduced a certifiably optimal algorithm that jointly estimates camera poses and per-image scaling factors that leverage pretrained monocular depth predictions.

**Feed-forward Scene Reconstruction**    The seminal work DUSt3R [73] has spawned multiple followup works in feed-forward scene reconstruction. DUSt3R takes in a pair of images and for each image, outputs a dense point map in the reference frame of the first camera. From the point maps, the camera focal lengths can be estimated using the Weiszfeld algorithm [51] and poses can be recovered using multiple methods such as a 3-point RANSAC [48, 40]. MASt3R follows a similar design but also outputs descriptors that can be used to generate pairwise correspondences between the two frames. MASt3R-SFM [15] demonstrates global optimization of multiple images using MASt3R but computation scales quickly with the number of frames.

To extend the idea of DUSt3R to multiple frames, Spann3R [70] leverages a learned memory module and Cut3R [72] uses a recurrent state model. Both can incrementally reconstruct a scene using multiple images, but are each limited to short sequences. Recently, Pow3R [29] extends the DUSt3R framework to optionally take in any estimates of any combination of camera intrinsics, poses, and depth (which may be sparse or dense) and demonstrates substantial improvement in scene reconstruction and pose estimation given the added inputs. Splatt3R [62] extends the DUSt3R idea to Gaussian Splatting [31] by directly outputting the Gaussian Splatting parameters given two views, and PreF3R [8] extends this to multiple views using a similar memory framework as Spann3R. Reloc3r [14] modifies the DUSt3R framework to directly output relative camera poses and uses motion averaging to recover absolute poses with respect to a map database.

Most similar to ours is MASt3R-SLAM [46]. MASt3R-SLAM leverages MASt3R to construct an impressive real time dense monocular SLAM system that does not require known calibration. Their pipeline also includes efficient optimization over $Sim(3)$ poses and loop closures. Since MASt3R is limited to two input frames at a time, here, we desire to build on top of the more powerful VGGT architecture for a SLAM system which can leverage broader information of the scene by taking in an arbitrary number of frames for feed-forward reconstruction (bounded by computational limits) and provides direct estimates of camera poses. However, as mentioned, fusing submaps from VGGT goes beyond a traditional point cloud registration problem as alignment cannot be effectively performed with only a similarity transformation. Unlike MASt3R-SLAM, as will be discussed in Sec. 4.2, we do not need to estimate correspondences between frames.

An alternative paradigm, scene coordinate regression, with works such as ACE [3] and DSAC* [4], estimates world points from images with respect to a global scene frame by using a scene specific trained network.

**Optimization over the Special Linear group**    To the best of our knowledge, we are the first work to create a factor graph optimization for point cloud alignment on the $SL(4)$ manifold. Prior works use optimization on the $SL(3)$ manifold (corresponding to the 8-DOF homography matrix commonly used in image alignment) for aligning multiple images for panoramic stitching [22, 61, 43, 42, 44, 36] and dense SLAM [35]. The 15-DOF homography matrix is used for classical tasks such as auto-calibration [23], and good practices for estimating homography are extensively studied in [24].

## 3 Review: VGGT

Here, we provide the relevant preliminaries of VGGT [71]. VGGT takes as input an image set $\mathcal{I} = \{\mathbf{M}_1, \cdots, \mathbf{M}_{\bar{w}}\}$, which consists of $\bar{w}$ images, tokenizes them with a fine-tuned DINO [49] backbone, and then applies Alternating-Attention (alternating between applying global and frame-wise attention). The output tokens can then be passed to a camera head to estimate intrinsics and camera poses (defined with respect to the first frame), or to Dense Prediction Transformer (DPT)

heads [54], which outputs dense depth maps for each image, a dense point map (where the points of each camera are defined with respect to the first camera), and dense features for point tracking, with confidence estimates provided for each.

In this paper, we use the dense depth maps $\mathcal{D} = \{\mathbf{D}_1, \cdots, \mathbf{D}_{\bar{w}}\}$ and confidence score maps $\mathcal{C} = \{\mathbf{C}_1, \cdots, \mathbf{C}_{\bar{w}}\}$ (as they are fully dense, the width and height of corresponding components in $\mathbf{M}$, $\mathbf{D}$, and $\mathbf{C}$ are the same). We refer to the outputs from each $\mathcal{I}$ as submap $\mathcal{S}$, which will correspond to a node in pose graph optimization for VGGT-SLAM. We do not run the 3D point DPT head as it was observed in Wang *et al.* [71] that more accurate point clouds can be achieved by inverse projecting $\mathbf{D}$ using projection matrices from the camera head, giving us a dense point cloud which is defined with respect to the coordinate frame of the first camera in $\mathcal{I}$. We denote this point cloud as $\mathbf{X}^{\mathcal{S}}$. To filter unreliable points, we prune points whose associated confidence values in the confidence maps are less than $\tau_{\mathrm{conf}}$ of the average confidence across $\mathcal{C}$.

## 4 VGGT-SLAM

Here we describe the design of our VGGT-SLAM system. In Sec. 4.1 we determine how to generate a list of images that will be passed to VGGT to produce a local submap, $\mathcal{S}$. In Sec. 4.2, we provide a discussion of projective ambiguity and show how we can align two overlapping submaps by estimating a relative 15-DOF homography matrix between sequential submaps, and in Sec. 4.3, we describe the process of adding loop closure constraints between non-sequential submaps. Finally, in Sec. 4.4 we show how we can globally optimize all submap alignments into a consistent map by optimizing on the $\mathrm{SL}(4)$ manifold.

### 4.1 Incremental submap-based keyframe selection and generation

First, we begin by describing how to incrementally construct submaps and organize keyframes within each submap from sequentially incoming images. For this, we construct an image set $\mathcal{I}_{\mathrm{latest}}$. As is typical in visual SLAM [57, 46, 64], we select an image as a keyframe if disparity (which we estimate using Lucas-Kanade [38]) with respect to the previous keyframe is larger than a user-defined threshold $\tau_{\mathrm{disparity}}$. Even though VGGT demonstrates monocular depth capabilities [71] from learned priors, utilizing images with sufficient disparity improves relative depth estimation performance as it adds multi-view information and additionally reduces the number of images to process.

If sufficiently high disparity is estimated, the current image is designated a keyframe and added to a list of images, $\mathcal{I}_{\mathrm{latest}}$, until the size of the list reaches a set limit $w$. In addition to $\mathcal{I}_{\mathrm{latest}}$, each submap's associated image set is constructed by concatenating two additional sets of images. The first set includes a single image chosen as the last non-loop-closure image from the previous submap, denoted as $\mathbf{M}_{\mathrm{prior}}$. Up to $w_{\mathrm{loop}}$ images to be used for loop closures (discussed in Sec. 4.3) may also be appended at the end of the collection, forming the final image set for the submap as $\mathcal{I}_{\mathrm{latest}} \leftarrow \{\mathbf{M}_{\mathrm{prior}}\} \cup \mathcal{I}_{\mathrm{latest}} \cup \mathcal{I}_{\mathrm{loop}}$. This image set is then passed to VGGT to generate the submap, $\mathcal{S}_{\mathrm{latest}}$.

### 4.2 Local submap alignment addressing projective ambiguity

Given two overlapping submaps $\mathcal{S}_i$ and $\mathcal{S}_j$ generated as described in Sec. 4.1, which have point clouds $\mathbf{X}^{\mathcal{S}_i}$ and $\mathbf{X}^{\mathcal{S}_j}$ in their respective local submap frames, our objective is to solve for a transformation, $\mathbf{H}_j^i \in \mathbb{R}^{4 \times 4}$ that aligns the two submaps such that for any noise-free corresponding points $\mathbf{X}_a^{\mathcal{S}_i}$, $\mathbf{X}_b^{\mathcal{S}_j} \in \mathbb{R}^3$, the following relation holds:

$$\mathbf{X}_a^{\mathcal{S}_i} = \mathbf{H}_j^i \mathbf{X}_b^{\mathcal{S}_j}, \tag{1}$$

where for simplicity, we overload notation such that $\mathbf{X}^{\mathcal{S}}$ is in homogeneous coordinates when multiplied by the $4 \times 4$ homography matrix. Under a typical 3D point cloud alignment problem, for example from LIDAR SLAM [33], $H$ would represent a translation and rotation in $\mathrm{SE}(3)$. If the point clouds additionally differ in scale, then $\mathbf{H}$ would be on $\mathrm{Sim}(3)$, the group of similarity transformations. However, here we do not have typical point clouds as $\mathbf{X}^{\mathcal{S}}$ is constructed by uncalibrated cameras. Thus, we recall *the Projective Reconstruction Theorem [26, Chapter 10.3], which in summary*

*states that if correspondences between two images from uncalibrated cameras uniquely determine the fundamental matrix, then the correspondences may be used to reconstruct the corresponding 3D points up to a 15-DOF homography transformation. This transform is the same for any such corresponding points, except those on the line connecting the camera centers as these points cannot be reconstructed uniquely.* Relevant to our setup, the Projective Reconstruction Theorem also applies to a reconstruction with more than two cameras [26]. Thus, in the most general case, the reconstruction computed using a set of uncalibrated cameras differs from a metrically correct reconstruction by a projective transformation (*i.e.,* homography) $\mathbf{H}$. The matrix $\mathbf{H}$ has 15 DOF and can be mapped uniquely to the *special linear group*, $\mathrm{SL}(4)$, by normalizing with the determinant. The $\mathrm{SL}(4)$ group consists of all real-valued $4 \times 4$ matrices with unit determinant. Note this is not the same as the more common 8 DOF homography matrix commonly used in planar computer vision tasks such as image warping, which belongs to $\mathrm{SL}(3)$. The reconstruction can be transformed to an affine reconstruction (*i.e.,* parallel lines are preserved) when scene priors are available, for example if points are known to lie on parallel lines. If further priors are known, such as lines in the scene are orthogonal, then the reconstruction can be converted to a metric reconstruction (differing by only a similarity transform to the true Euclidean reconstruction). VGGT is thus able to leverage learned scene priors to potentially estimate metric reconstruction, but as we have shown in Fig. 1, in the most general case when estimates of scene priors are unreliable, the reconstruction differs by a projective ambiguity, requiring a 15-DOF homography matrix to rectify. We will now estimate such a homography.

By our construction of the submaps that they share a same image, we have an atypical advantage in solving for $\mathbf{H}$ as we have a dense set of correspondences without needing to estimate associations.

As is well known by the direct reconstruction method [26], the optimal homography in (1) can be solved in closed form as a solution to the following homogeneous linear system:

$$\mathbf{A}_k \mathbf{h} = 0 \tag{2}$$

with $\mathbf{h} \in \mathbb{R}^{16}$ containing the flattened parameters of the homography and $\mathbf{A}_k$ contains constraints for a particular pair of 3D points. A minimum solution requires 5 points (*i.e.,* $k \in \{1 : 5\}$), and to build in robustness to incorrect depth measurements from VGGT, we solve (2) using RANSAC [18] with a 5-point solver. As the homography matrix is estimated up to scale, we scale by the fourth root of the determinant such that the determinant is unity and the resulting matrix belongs to $\mathrm{SL}(4)$.

**Transformation of camera poses via homography**   Using a homography between reference frames $i$ and $j$, $\mathbf{H}_j^i$, the camera poses can be corrected using the following [26]: $\mathbf{P}_i = \mathbf{P}_j \mathbf{H}_j^{i\,-1}$, where $\mathbf{P} \in \mathbb{R}^{3 \times 4}$ is the camera matrix created from the poses and intrinsic estimates from VGGT. We can then decompose $\mathbf{P}$ to recover the camera pose.

### 4.3   Loop closures

Our procedure for creating loop closures for VGGT-SLAM consists of two steps: (i) performing image retrieval (*i.e.,* setting $\mathcal{I}_{\mathrm{loop}}$ in Sec. 4.1), and (ii) estimating relative homographies, which are then added to the factor graph as loop closure constraints (Sec. 4.4). First, for image retrieval, when constructing a submap, we compute and store an image descriptor for each keyframe using SALAD [28]. Then, once $\mathcal{I}_{\mathrm{latest}}$ reaches its size threshold $w$, we search over the image descriptors in the previous submaps $\mathcal{S}_i \ \forall i \in \{1 : \mathrm{lastest} - \tau_{\mathrm{interval}}\}$ to fetch a set of frames of size $w_{\mathrm{loop}}$ that have the highest similarity (using the L2 norm) to any of the keyframes in $\mathcal{I}_{\mathrm{latest}}$, and also exceed a user-defined similarity threshold $\tau_{\mathrm{desc}}$ to reduce false positive matches. These frames make up $\mathcal{I}_{\mathrm{loop}}$, which is added to the list of keyframes for the current submap, and then all frames are sent to VGGT as described in Sec. 4.1.

Next, given the estimated submap, $\mathcal{S}_{\mathrm{latest}}$, from VGGT, we estimate the relative homographies between the loop closure frames in $\mathcal{S}_{\mathrm{latest}}$ and the submaps, $\mathcal{S}_i$, retrieved during the image retrieval process described above. As in Sec. 4.2, we again have the benefit of not requiring an estimate of correspondences to compute homographies for loop closures; thus, we can directly use (2) between the frames in $\mathcal{I}_{\mathrm{loop}}$ and their respective identical frames in the submap where they originated. This then provides $w_{\mathrm{loop}}$ loop closure constraints between $\mathcal{S}_{\mathrm{latest}}$ and the corresponding submaps.

Note that a potential alternative is to get the descriptor using the output tokens from VGGT's fine-tuned DINO backbone. This alleviates using a separate descriptor module and storing the physical images. However, this requires storing larger features in memory compared to the relatively small SALAD features, and the system memory needed to store the images in our base approach is relatively low.

### 4.4 Backend: Nonlinear factor graph optimization on the $\mathrm{SL}(4)$ manifold

Given all relative homographies $\mathbf{H}_j^i$ between submaps $\mathcal{S}_i$ and $\mathcal{S}_j$, our goal is to compute the absolute homographies $\mathbf{H}_i$ that transform all submaps into a common global reconstruction. To achieve this, we formulate a nonlinear factor graph optimization problem[2] based on Maximum A Posteriori (MAP) estimation [2, 17, 20]. Specifically, we estimate the absolute homographies by minimizing the following cost function under Gaussian noise assumptions on the relative homographies:

$$\hat{\mathcal{H}} = \underset{\mathbf{H} \in \mathrm{SL}(4)}{\arg\min} \sum_{(i,j) \in \mathcal{L}} \left\| \mathrm{Log}\left( \mathbf{H}_i^{-1} \mathbf{H}_j \left( \mathbf{H}_j^i \right)^{-1} \right) \right\|_{\Omega_{ij}^{\mathbf{H}}}^2, \tag{3}$$

where $\mathrm{Log}(\cdot)$ is the mapping function that transforms a group element to a (vectorized) element of the corresponding Lie algebra, $\mathcal{L}$ denotes an index set of constraints that includes odometry and loop closures, and we set $\Omega_{ij}^{\mathbf{H}} \in \mathbb{R}^{15 \times 15}$ to the identity matrix.

To solve (3), we iteratively compute state increments by solving a linearized least squares problem. To this end, we define $\boldsymbol{\xi} \in \mathbb{R}^{15}$ as the tangent-space parameterization of $\mathrm{SL}(4)$, the mapping function $\mathrm{Exp} : \mathbb{R}^{15} \rightarrow \mathrm{SL}(4)$, which satisfies $\mathrm{Log}\big(\mathrm{Exp}\left(\boldsymbol{\xi}\right)\big) = \boldsymbol{\xi}$ and $\mathrm{Exp}\left(\boldsymbol{\xi}\right) = \exp(\boldsymbol{\xi}^\wedge) = \mathbf{H}$. In particular, $\boldsymbol{\xi}^\wedge$ is a Lie algebra element of $\mathfrak{sl}(4)$, computed by summing the $k$-th component of $\boldsymbol{\xi}$ with its $k$-th corresponding generator $\mathbf{G}_k \; \forall k : \{1 : 15\}$ (i.e., $\boldsymbol{\xi}^\wedge = \sum_{k=1}^{15} \boldsymbol{\xi}_k \, \mathbf{G}_k$) [16]. More details can be found in Appendix A.

Next, defining the measurement function as $h\big(\boldsymbol{\xi}_i, \boldsymbol{\xi}_j\big) = \mathrm{Log}\left(\mathbf{H}_i^{-1}\mathbf{H}_j\right)$, the incremental update of each pose can be approximated using Taylor's expansion [13] as follows:

$$h\big(\boldsymbol{\xi}_i \oplus \boldsymbol{\delta}_i, \boldsymbol{\xi}_j \oplus \boldsymbol{\delta}_j\big) \simeq h\big(\boldsymbol{\xi}_i, \boldsymbol{\xi}_j\big) \oplus \{\mathbf{J}_i \boldsymbol{\delta}_i + \mathbf{J}_j \boldsymbol{\delta}_j\}, \; \boldsymbol{\xi} \oplus \boldsymbol{\delta} = \mathbf{H}\,\mathrm{Exp}\left(\boldsymbol{\delta}\right) \tag{4}$$

where $\mathbf{J}_i = -\mathrm{Ad}_{\mathbf{H}_i^{-1}\mathbf{H}_j}$ and $\mathbf{J}_j = I_{15 \times 15}$. Here, $\mathrm{Ad}_{\mathbf{H}}$ is the adjoint map, defined as $\mathrm{Ad}_{\mathbf{H}} = \mathbf{B}^{-1}\mathbf{H} \otimes \mathbf{H}^{-\mathsf{T}}\mathbf{B}$ [16], where $\mathbf{B} = [\mathrm{vec}\left(\mathbf{G}_1\right) \; \mathrm{vec}\left(\mathbf{G}_2\right) \; \cdots \; \mathrm{vec}\left(\mathbf{G}_{15}\right)] \in \mathbb{R}^{16 \times 15}$ in the case of the $\mathrm{SL}(4)$ manifold [9], and $\otimes$ denotes the Kronecker product, which forms a block matrix by multiplying each element of the first matrix with the entire second matrix.

Finally, we can formulate the linearized residuals and the resulting local problem at the linearization point $\mathbf{H}_j^i$ as follows:

$$\hat{\mathcal{D}} = \underset{\boldsymbol{\delta} \in \mathcal{D}}{\arg\min} \sum_{(i,j) \in \mathcal{L}} \|\boldsymbol{e}_{ij} + \mathbf{J}_i \boldsymbol{\delta}_i + \mathbf{J}_j \boldsymbol{\delta}_j\|_{\Omega_{ij}^{\mathbf{H}}}^2, \; \boldsymbol{e}_{ij} = \mathrm{Log}\left( \mathbf{H}_i^{-1} \mathbf{H}_j \left( \mathbf{H}_j^i \right)^{-1} \right). \tag{5}$$

To solve (5), we use the Levenberg-Marquardt optimizer [55], and at each iteration, the poses are updated on the Lie group as $\mathbf{H} \leftarrow \mathbf{H}\,\mathrm{Exp}(\hat{\boldsymbol{\delta}})$ [63].

## 5 Experiments

We follow similar experiments as MASt3R-SLAM to evaluate camera pose estimation and dense reconstruction in Sec. 5.2 and Sec. 5.3 respectively, demonstrate qualitative results in Sec. 5.5, and finally perform ablations in Sec. 5.6.

---

[2]In our case, this is an extension of pose graph optimization, where we estimate absolute poses from pairwise pose measurements.

## 5.1 Experimental setup

We evaluate VGGT-SLAM on standard RGB SLAM benchmarks to assess both camera pose estimation accuracy and dense mapping quality. For evaluation of pose estimation, we employ the 7-Scenes [60] and TUM RGB-D [65] datasets, and report root mean square error (RMSE) of the absolute trajectory error (ATE) using evo [21]. Since 7-Scenes [60] provides scene ground truth, this dataset is also used to evaluate dense mapping quality in terms of *accuracy*, *completion*, and *Chamfer distance* [46].

As baseline approaches, we primarily compare VGGT-SLAM with DROID-SLAM [67] and MASt3R-SLAM [46] as the state-of-the-art learning-based SLAM approaches in uncalibrated scenarios (and Spann3R [70] for dense evaluation). We use reported numbers from MASt3R-SLAM [46] for baselines, except for the uncalibrated version of DROID-SLAM. Although DROID-SLAM requires camera intrinsics, we also evaluate it in an uncalibrated setting by estimating intrinsics with an automatic calibration pipeline [69], as is suggested by Murai *et al.* [46]. While our approach operates without camera calibration, we also include comparison with state-of-the-art methods [7, 34, 79, 66, 12, 82] provided with camera intrinsics. Due to potential randomness in our approach caused by RANSAC, we report the average performance over five runs, which have a low spread (small standard deviation) as shown in Sec. 5.6.

We refer to a simpler $\mathrm{Sim}(3)$ version of VGGT-SLAM as Ours ($\mathrm{Sim}(3)$), for which we follow similar structure as our $\mathrm{SL}(4)$ pipeline except we align relative rotation and translation between submaps using pose estimates from VGGT and estimate a scale correction by comparing the estimated point clouds of the overlapping images. Loop closures and relative factors are added to the factor graph as $\mathrm{SE}(3)$ factors.

We use an NVIDIA GeForce RTX 4090 GPU with AMD Ryzen Threadripper 7960X CPU. For parameters, we set $w_{\mathrm{loop}} = 1$, $\tau_{\mathrm{disparity}} = 50$ pixels, $\tau_{\mathrm{interval}} = 2$, $\tau_{\mathrm{desc}} = 0.8$, and $\tau_{\mathrm{conf}} = 25\%$. We also use 300 RANSAC iterations with a threshold of 0.01. We show evaluations of both the $\mathrm{SL}(4)$ and $\mathrm{Sim}(3)$ version of VGGT-SLAM with different submap sizes (*i.e.,* different values for $w$).

## 5.2 Pose estimation evaluation

As shown in Tables 1 and 2, VGGT-SLAM performs comparable to the top performing uncalibrated baselines on 7-Scenes and TUM RGB-D. On 7-Scenes for instance, VGGT-SLAM has approximately the same average APE as the top performing baseline MASt3R-SLAM. On the TUM dataset, the $\mathrm{SL}(4)$ version of VGGT-SLAM performs the best overall with an average error of 0.053 m. This demonstrates that we are able to extend VGGT to multiple sequences while introducing a new category of SLAM system by optimizing submap alignment as $\mathrm{SL}(4)$ factors. Here, we observe that our $\mathrm{Sim}(3)$ version also performs well, as these scenes are generally cases where VGGT is able to leverage strong priors for metric reconstruction. Thus, while we have shown cases where $\mathrm{SL}(4)$ is needed (Fig. 1), the addition of higher degrees of freedom with our novel SLAM formulation maintains competitive performance, while improving some more challenging cases.

One particular scene where our method underperforms is on the TUM `floor` scene. This highlights a challenge of estimating homography, which is the presence of degeneracy in the case of a planar scene. The floor scene contains several images that only view the flat floor leading to non-unique solutions for the homography matrix, which causes the overall reconstruction to diverge. Building robustness for the planar case is an important component for $\mathrm{SL}(4)$ SLAM, which we leave as an exciting direction for future work. The TUM `360` scene is particularly challenging for smaller submap sizes (although handled well with $w = 32$) because smaller submaps are more likely to encounter approximately pure rotation in this scene, which can have reduced depth accuracy and hence a higher outlier ratio when running 5-point RANSAC to estimate homography.

Table 1: Root mean square error (RMSE) of absolute trajectory error (ATE) on 7-Scenes [60] (unit: m). The gray rows indicate the results using the calibrated camera intrinsics and the * symbol indicates that the baseline is evaluated in the uncalibrated mode. Green is best and light green is second best.

| | Method | Sequence | | | | | | | Avg |
|---|---|---|---|---|---|---|---|---|---|
| | | chess | fire | heads | office | pumpkin | kitchen | stairs | |
| Calib. | NICER-SLAM [82] | **0.033** | 0.069 | 0.042 | 0.108 | 0.200 | **0.039** | 0.108 | 0.086 |
| | DROID-SLAM [67] | 0.036 | 0.027 | 0.025 | **0.066** | 0.127 | 0.040 | 0.026 | 0.049 |
| | MASt3R-SLAM [46] | 0.053 | **0.025** | **0.015** | 0.097 | **0.088** | 0.041 | **0.011** | **0.047** |
| Uncalib. | DROID-SLAM* [67] | 0.047 | 0.038 | 0.034 | 0.136 | 0.166 | 0.080 | 0.044 | 0.078 |
| | MASt3R-SLAM* [46] | 0.063 | 0.046 | 0.029 | 0.103 | 0.114 | 0.074 | 0.032 | 0.066 |
| | Ours $(\mathrm{Sim}(3), w = 32)$ | 0.037 | 0.026 | 0.018 | 0.104 | 0.133 | 0.061 | 0.093 | 0.067 |
| | Ours $(\mathrm{SL}(4), w = 32)$ | 0.036 | 0.028 | 0.018 | 0.103 | 0.133 | 0.058 | 0.093 | 0.067 |

Table 2: Root mean square error (RMSE) of absolute trajectory error (ATE) on TUM RGB-D [65] (unit: m). The gray rows indicate the results using the calibrated camera intrinsics and the * symbol indicates that the baseline is evaluated in the uncalibrated mode. Green is best and light green is second best.

| | Method | Sequence | | | | | | | | | Avg |
|---|---|---|---|---|---|---|---|---|---|---|---|
| | | 360 | desk | desk2 | floor | plant | room | rpy | teddy | xyz | |
| Calib. | ORB-SLAM3 [7] | × | 0.017 | 0.210 | × | 0.034 | × | × | × | **0.009** | N/A |
| | DeepV2D [66] | 0.243 | 0.166 | 0.379 | 1.653 | 0.203 | 0.246 | 0.105 | 0.316 | 0.064 | 0.375 |
| | DeepFactors [12] | 0.159 | 0.170 | 0.253 | 0.169 | 0.305 | 0.364 | 0.043 | 0.601 | 0.035 | 0.233 |
| | DPV-SLAM [34] | 0.112 | 0.018 | 0.029 | 0.057 | 0.021 | 0.330 | 0.030 | 0.084 | 0.010 | 0.076 |
| | DPV-SLAM++ [34] | 0.132 | 0.018 | 0.029 | 0.050 | 0.022 | 0.096 | 0.032 | 0.098 | 0.010 | 0.054 |
| | GO-SLAM [79] | 0.089 | **0.016** | 0.028 | 0.025 | 0.026 | 0.052 | **0.019** | 0.048 | 0.010 | 0.035 |
| | DROID-SLAM [67] | 0.111 | 0.018 | 0.042 | **0.021** | **0.016** | **0.049** | 0.026 | 0.048 | 0.012 | 0.038 |
| | MASt3R-SLAM [46] | **0.049** | **0.016** | **0.024** | 0.025 | 0.020 | 0.061 | 0.027 | **0.041** | **0.009** | **0.030** |
| Uncalib. | DROID-SLAM* [67] | 0.202 | 0.032 | 0.091 | 0.064 | 0.045 | 0.918 | 0.056 | 0.045 | 0.012 | 0.158 |
| | MASt3R-SLAM* [46] | 0.070 | 0.035 | 0.055 | 0.056 | 0.035 | 0.118 | 0.041 | 0.114 | 0.020 | 0.060 |
| | Ours $(\mathrm{Sim}(3), w = 32)$ | 0.123 | 0.040 | 0.055 | 0.254 | 0.022 | 0.088 | 0.041 | 0.032 | 0.016 | 0.074 |
| | Ours $(\mathrm{SL}(4), w = 32)$ | 0.071 | 0.025 | 0.040 | 0.141 | 0.023 | 0.102 | 0.030 | 0.034 | 0.014 | 0.053 |

## 5.3 Dense reconstruction evaluation

Following the protocol of MASt3R-SLAM, we provide dense reconstruction performance on 7-Scenes; see Table 3. Here, we observe that while performance is comparable across methods, VGGT-SLAM achieves the best performing accuracy and Chamfer distance, demonstrating the high accuracy of our dense point cloud reconstruction.

## 5.4 Timing Analysis

Table 3: Root mean square error (RMSE) reconstruction evaluation on 7-Scenes [60] (unit: m). @$n$ indicates a keyframe every $n$ images.

| | Method | 7-Scenes | | | |
|---|---|---|---|---|---|
| | | ATE↓ | Acc.↓ | Complet.↓ | Chamfer↓ |
| Calib. | DROID-SLAM [67] | 0.049 | 0.141 | 0.048 | 0.094 |
| | MASt3R-SLAM [46] | **0.047** | 0.089 | 0.085 | 0.087 |
| | Spann3R @20 [70] | N/A | 0.069 | 0.047 | 0.058 |
| | Spann3R @2 [70] | N/A | 0.124 | **0.043** | 0.084 |
| Uncalib. | MASt3R-SLAM* [46] | 0.066 | 0.068 | 0.045 | 0.056 |
| | Ours $(\mathrm{Sim}(3), w = 32)$ | 0.067 | **0.052** | 0.062 | 0.057 |
| | Ours $(\mathrm{SL}(4), w = 32)$ | 0.067 | **0.052** | 0.058 | **0.055** |

In Table 4 we show approximate timing results comparing the primary components of the $\mathrm{Sim}(3)$ and $\mathrm{SL}(4)$ versions of VGGT-SLAM. As expected, the total time used to run optical flow and determine keyframes for all frames in the submap (keyframe detection), the total time to run SALAD and perform image retrieval for all frames in the submap (loop closure detection), and the time to run the VGGT model (VGGT inference) is identical up to random variations. The time to optimize the factor graph using GTSAM (backend optimization) is very fast, taking only about half a millisecond for both variants, since the factor graph in VGGT-SLAM is relatively small compared to factor graphs used in typical real time visual odometry problems [56]. The primary runtime difference between the $\mathrm{Sim}(3)$ and $\mathrm{SL}(4)$ variants is the time to compute relative edge constraints (relative transformation estimation) since the 5-point RANSAC estimation takes about 17 ms longer than the time to compute the $\mathrm{Sim}(3)$ transformation. However, this increase is comparatively small, only about 2.5 percent of the time needed to run VGGT.

Memory needed to run VGGT for VGGT-SLAM is independent of the total number of images in the scene since VGGT is run with fixed size submaps. An analysis of memory usage for running VGGT with varying numbers of images can be found in [10].

Table 4: Runtime comparison between $\mathrm{Sim}(3)$ and $\mathrm{SL}(4)$ versions of VGGT-SLAM on the `office_loop` sequence with window size $w = 16$ (unit: msec). Runtime is averaged over five runs.

| Stage | VGGT-SLAM w/ $\mathrm{Sim}(3)$ | VGGT-SLAM w/ $\mathrm{SL}(4)$ |
|---|---|---|
| Keyframe detection | 176 ms | 176 ms |
| VGGT inference | 662 ms | 662 ms |
| Loop closure detection | 105 ms | 105 ms |
| Relative transformation estimation | 11 ms | 28 ms |
| Backend optimization | 0.4 ms | 0.5 ms |

## 5.5 Qualitative results

We present qualitative results to illustrate the mapping fidelity of VGGT-SLAM using $\mathrm{SL}(4)$ optimization. In Fig. 2, we show an example reconstruction from the office scene in 7-Scenes and from a longer 55 meter trajectory that loops inside an office corridor. In addition to the dense reconstruction, we also show all mapped camera poses, where different colors indicate the submap associated with each image. In particular, the office corridor loop clearly shows 22 different submaps which have been joined into a globally consistent map with a loop closure at the end of the trajectory.

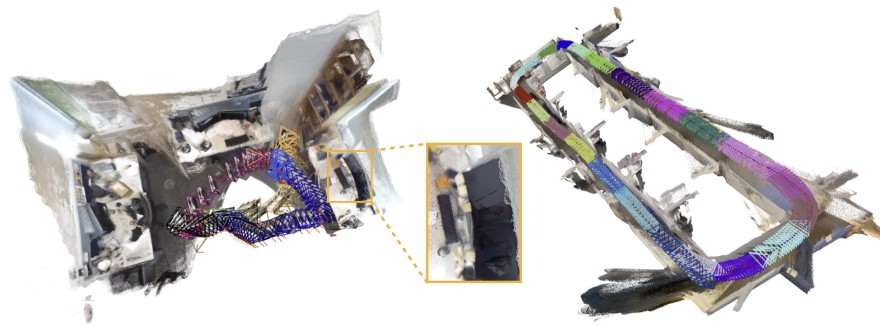

Figure 2: Reconstruction and pose estimates from VGGT-SLAM on the office scene from 7-Scenes showing 8 submaps and from a custom scene showing a 55 meter loop around an office corridor with 22 submaps. Both use $w = 16$. Different image colors indicate the submap associated with each image.

In Fig. 1, we show two select examples where using only $\mathrm{Sim}(3)$ is unable to align overlapping submaps while our $\mathrm{SL}(4)$ alignment strategy is able to rectify the projective ambiguity between submaps. Thus, while we have shown that $\mathrm{Sim}(3)$ generally achieves accurate performance across our quantitative experiments, in the general case where a feed-forward reconstruction method like VGGT is unable to estimate a metric reconstruction (for reasons discussed in Sec. 4.2) due to the computational limits, our introduction of an $\mathrm{SL}(4)$-based SLAM system shows promise in leveraging the potential of a high accuracy, dense, learning-based SLAM system. For Fig. 1, $\tau_{\mathrm{disparity}}$ is set to 0 to highlight the impact of projective ambiguity, which degrades the performance of $\mathrm{Sim}(3)$ alignment and affects overall map quality.

## 5.6 Ablations

In Fig. 3 we provide multiple ablation studies which show the following for three different submap sizes ($w = 8, 16, 32$): (a) improved pose accuracy of VGGT-SLAM when loop closures are leveraged along with showing a tight statistical spread of results from averaging 5 runs from our experiments, (b) that loop closures generally lead to increasing reduction in pose error as the number of submaps increases since there are an increased number of loop closures, (c) the effect of different values of $\tau_{\mathrm{conf}}$, which as expected, larger values lead to higher accuracy on dense reconstruction and lower completion, with our default value of 25% showing an appropriate balance.

## 6 Limitations

We have presented a new type of SLAM system that addresses the issue of projective ambiguity from an underlying feed-forward scene reconstruction method (in our case, VGGT). As creating a factor

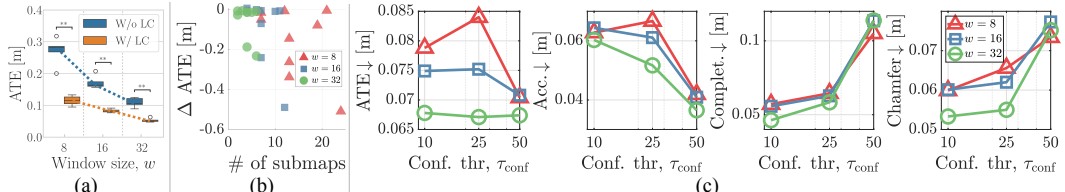

Figure 3: Ablation studies: (a) Effect of loop closure (LC) on absolute trajectory error (ATE) across different window sizes, $w$, in TUM [65], where ** annotations indicate measurements with $10^{-3} < p\text{-value} \leq 10^{-2}$ after a paired $t$-test. (b) Reduction in ATE achieved by incorporating loop closure in TUM, highlighting that as the number of submaps increases, our $\mathrm{SL}(4)$-based optimization leads to greater reductions in ATE. (c) Performance changes with respect to the confidence threshold $\tau_{\mathrm{conf}}$ in ATE, accuracy, completion, and Chamfer distance under varying window sizes in 7-Scenes [60].

graph that optimizes on the $\mathrm{SL}(4)$ manifold is a new paradigm for the SLAM problem, it leaves much ground for further improvements. In particular, the estimation of the full 15-DOF homography matrix is degenerate in the case of planar points, which can lead to unstable solutions as we have observed in the planar floor scene of the TUM dataset. Our current implementation of homography using points from VGGT is also vulnerable to outliers. While we use a 5-point RANSAC to reduce this issue, the presence of a high outlier ratios or adversarial outliers (which are present due to local consistency of points in VGGT) can cause incorrect homography estimates as discussed in Sec. 5.2. The ray-based matching in MASt3R-SLAM provides robustness to errors in depth measurements, and a similar method can potentially be adapted for homography estimation. Additionally, 15 DOF give rise to added opportunity of scene drift. While our addition of loop closures substantially corrects drift, an inaccurate relative homography estimate or long time between loop closures can cause not just drift in scale, rotation, and translation seen in classical SLAM, but also in scene perspective, which opens up an interesting area of research into further optimization into $\mathrm{SL}(4)$-based SLAM. Finally, lens distortion is not rectified by the homography matrix since a projective transformation preserves straight lines, and thus images are assumed to be undistorted when running VGGT-SLAM.

## 7    Conclusion

In this study, we have leveraged VGGT, a feed-forward reconstruction model, to incrementally construct a dense map from uncalibrated monocular cameras, proposing a novel SLAM framework called *VGGT-SLAM*, which locally and globally (through loop closures) aligns submaps from VGGT. By exploring VGGT's geometric understanding through the lens of classical multi-view computer vision, we have shown that in the general case, these submaps must be aligned with a projective transformation, and in doing so we have created the first factor graph SLAM system optimized on the $\mathrm{SL}(4)$ manifold. In future work we will further investigate conditions under which $\mathrm{Sim}(3)$ optimization suffices and investigate techniques to actively employ both $\mathrm{Sim}(3)$ and $\mathrm{SL}(4)$ optimization in a unified system to enable a more robust SLAM system for real-time performance.

## Acknowledgments

This work is supported in part by the NSF Graduate Research Fellowship Program under Grant 2141064, the ONR RAPID program, and the National Research Foundation of Korea (NRF) grant funded by the Korea government (MSIT) (No. RS-2024-00461409). The authors would like to gratefully acknowledge Riku Murai for assisting us with benchmarking.

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

# A  Tangent space of $\mathrm{SL}(4)$

Here, we provide the explicit 15 generators, $\mathbf{G}_k \; \forall k : \{1 : 15\}$, of $\mathrm{SL}(4)$, which allow us to relate the Lie algebra $\mathfrak{sl}(4)$ to the Lie group $\mathrm{SL}(4)$.

The tangent space of $\mathrm{SL}(4)$ consists of all $4 \times 4$ real matrices with zero trace. Thus, there are 15 generators, $\mathbf{G}_k$, where 12 of them are defined as $\boldsymbol{E}_{ab}$ for $a \neq b$ where 1 is in the $(a, b)$ entry and 0, elsewhere. The remaining three generators are $\boldsymbol{B}_1 = \mathrm{diag}(1, -1, 0, 0)$, $\boldsymbol{B}_2 = \mathrm{diag}(0, 1, -1, 0)$, $\boldsymbol{B}_3 = \mathrm{diag}(0, 0, 1, -1)$. Explicitly, the generators are as follows:

$$
\mathbf{G}_1 = \boldsymbol{E}_{01} = \begin{pmatrix} 0 & 1 & 0 & 0 \\ 0 & 0 & 0 & 0 \\ 0 & 0 & 0 & 0 \\ 0 & 0 & 0 & 0 \end{pmatrix}, \quad
\mathbf{G}_2 = \boldsymbol{E}_{02} = \begin{pmatrix} 0 & 0 & 1 & 0 \\ 0 & 0 & 0 & 0 \\ 0 & 0 & 0 & 0 \\ 0 & 0 & 0 & 0 \end{pmatrix}, \quad
\mathbf{G}_3 = \boldsymbol{E}_{03} = \begin{pmatrix} 0 & 0 & 0 & 1 \\ 0 & 0 & 0 & 0 \\ 0 & 0 & 0 & 0 \\ 0 & 0 & 0 & 0 \end{pmatrix},
$$

$$
\mathbf{G}_4 = \boldsymbol{E}_{10} = \begin{pmatrix} 0 & 0 & 0 & 0 \\ 1 & 0 & 0 & 0 \\ 0 & 0 & 0 & 0 \\ 0 & 0 & 0 & 0 \end{pmatrix}, \quad
\mathbf{G}_5 = \boldsymbol{E}_{12} = \begin{pmatrix} 0 & 0 & 0 & 0 \\ 0 & 0 & 1 & 0 \\ 0 & 0 & 0 & 0 \\ 0 & 0 & 0 & 0 \end{pmatrix}, \quad
\mathbf{G}_6 = \boldsymbol{E}_{13} = \begin{pmatrix} 0 & 0 & 0 & 0 \\ 0 & 0 & 0 & 1 \\ 0 & 0 & 0 & 0 \\ 0 & 0 & 0 & 0 \end{pmatrix},
$$

$$
\mathbf{G}_7 = \boldsymbol{E}_{20} = \begin{pmatrix} 0 & 0 & 0 & 0 \\ 0 & 0 & 0 & 0 \\ 1 & 0 & 0 & 0 \\ 0 & 0 & 0 & 0 \end{pmatrix}, \quad
\mathbf{G}_8 = \boldsymbol{E}_{21} = \begin{pmatrix} 0 & 0 & 0 & 0 \\ 0 & 0 & 0 & 0 \\ 0 & 1 & 0 & 0 \\ 0 & 0 & 0 & 0 \end{pmatrix}, \quad
\mathbf{G}_9 = \boldsymbol{E}_{23} = \begin{pmatrix} 0 & 0 & 0 & 0 \\ 0 & 0 & 0 & 0 \\ 0 & 0 & 0 & 1 \\ 0 & 0 & 0 & 0 \end{pmatrix},
$$

$$
\mathbf{G}_{10} = \boldsymbol{E}_{30} = \begin{pmatrix} 0 & 0 & 0 & 0 \\ 0 & 0 & 0 & 0 \\ 0 & 0 & 0 & 0 \\ 1 & 0 & 0 & 0 \end{pmatrix}, \quad
\mathbf{G}_{11} = \boldsymbol{E}_{31} = \begin{pmatrix} 0 & 0 & 0 & 0 \\ 0 & 0 & 0 & 0 \\ 0 & 0 & 0 & 0 \\ 0 & 1 & 0 & 0 \end{pmatrix}, \quad
\mathbf{G}_{12} = \boldsymbol{E}_{32} = \begin{pmatrix} 0 & 0 & 0 & 0 \\ 0 & 0 & 0 & 0 \\ 0 & 0 & 0 & 0 \\ 0 & 0 & 1 & 0 \end{pmatrix},
$$

$$
\mathbf{G}_{13} = \boldsymbol{B}_1 = \begin{pmatrix} 1 & 0 & 0 & 0 \\ 0 & -1 & 0 & 0 \\ 0 & 0 & 0 & 0 \\ 0 & 0 & 0 & 0 \end{pmatrix}, \quad
\mathbf{G}_{14} = \boldsymbol{B}_2 = \begin{pmatrix} 0 & 0 & 0 & 0 \\ 0 & 1 & 0 & 0 \\ 0 & 0 & -1 & 0 \\ 0 & 0 & 0 & 0 \end{pmatrix}, \quad
\mathbf{G}_{15} = \boldsymbol{B}_3 = \begin{pmatrix} 0 & 0 & 0 & 0 \\ 0 & 0 & 0 & 0 \\ 0 & 0 & 1 & 0 \\ 0 & 0 & 0 & -1 \end{pmatrix}.
$$

Thus, as briefly explained in Sec. 4.4, the relation between the Lie algebra, $\boldsymbol{\xi}^\wedge \in \mathfrak{sl}(4)$, and the Lie group $\mathbf{H} \in \mathrm{SL}(4)$ is given by:

$$
\mathbf{H} = \exp\left(\boldsymbol{\xi}^\wedge\right) = \exp\left(\sum_{k=1}^{15} \boldsymbol{\xi}_k \, \mathbf{G}_k\right). \tag{6}
$$

# B  Extra Quantitative Results

We provide addition results of evaluating on the 7-Scenes [60] and TUM RGB-D [65] datasets where we experiment with different submap sizes (Appendix B.1) and show the number of submaps and loop closures per scene (Appendix B.2).

## B.1  Evaluation with different submap sizes

Here we show results for the 7-Scenes and TUM RGB-D datasets in Tables 5 and 6 with different submap sizes ($w = 8, 16, 32$). For 7-Scenes, we also include results for $w = 1$. Recall that $w$ is the size of new images in the submap, so in the case of $w = 1$, each submap has one new image, one image from the prior submap, and up to one extra image from loop closures. For small submap size of $w = 1$, the backend becomes numerically unstable for some TUM scenes (consistently floor and 360) preventing an estimated alignment, and thus we do not include the $w = 1$ for TUM. This is due to reasons discussed in Sec. 6. Particularly, for the floor scene there are a large portion of images which only view a planar scene which makes the estimation of the full 15-DOF homography matrix

degenerate, and for the 360 scene, using a small submap size such as $w = 1$ is likely to encounter a pure rotation which can result in less accurate depth measurements from VGGT and hence reduced accuracy in our estimate of relative homography.

Table 5: Root mean square error (RMSE) of absolute trajectory error (ATE) on 7-Scenes [60] (unit: m). The * symbol indicates that the baseline is evaluated in the uncalibrated mode, all VGGT-SLAM configurations are evaluated in uncalibrated mode. Green is best and light green is second best.

| | Method | Sequence | | | | | | | Avg |
|---|---|---|---|---|---|---|---|---|---|
| | | chess | fire | heads | office | pumpkin | kitchen | stairs | |
| Uncalib. | DROID-SLAM* [67] | 0.047 | 0.038 | 0.034 | 0.136 | 0.166 | 0.080 | 0.044 | 0.078 |
| | MASt3R-SLAM* [46] | 0.063 | 0.046 | 0.029 | 0.103 | 0.114 | 0.074 | 0.032 | 0.066 |
| | Ours (Sim(3), $w = 1$) | 0.047 | 0.025 | 0.032 | 0.113 | 0.138 | 0.050 | 0.083 | 0.070 |
| | Ours (Sim(3), $w = 8$) | 0.039 | 0.027 | 0.020 | 0.108 | 0.144 | 0.053 | 0.080 | 0.067 |
| | Ours (Sim(3), $w = 16$) | 0.037 | 0.027 | 0.021 | 0.107 | 0.135 | 0.051 | 0.093 | 0.067 |
| | Ours (Sim(3), $w = 32$) | 0.037 | 0.026 | 0.018 | 0.104 | 0.133 | 0.061 | 0.093 | 0.067 |
| | Ours (SL(4), $w = 1$) | 0.089 | 0.046 | 0.072 | 0.119 | 0.147 | 0.055 | 0.100 | 0.090 |
| | Ours (SL(4), $w = 8$) | 0.041 | 0.060 | 0.043 | 0.106 | 0.206 | 0.054 | 0.078 | 0.084 |
| | Ours (SL(4), $w = 16$) | 0.036 | 0.065 | 0.037 | 0.107 | 0.139 | 0.050 | 0.093 | 0.075 |
| | Ours (SL(4), $w = 32$) | 0.036 | 0.028 | 0.018 | 0.103 | 0.133 | 0.058 | 0.093 | 0.067 |

Table 6: Root mean square error (RMSE) of absolute trajectory error (ATE) on TUM RGB-D [65] (unit: m). The * symbol indicates that the baseline is evaluated in the uncalibrated mode, all VGGT-SLAM configurations are evaluated in uncalibrated mode. Green is best and light green is second best.

| | Method | Sequence | | | | | | | | | Avg |
|---|---|---|---|---|---|---|---|---|---|---|---|
| | | 360 | desk | desk2 | floor | plant | room | rpy | teddy | xyz | |
| Uncalib. | DROID-SLAM* [67] | 0.202 | 0.032 | 0.091 | 0.064 | 0.045 | 0.918 | 0.056 | 0.045 | 0.012 | 0.158 |
| | MASt3R-SLAM* [46] | 0.070 | 0.035 | 0.055 | 0.056 | 0.035 | 0.118 | 0.041 | 0.114 | 0.020 | 0.060 |
| | Ours (Sim(3), $w = 8$) | 0.070 | 0.026 | 0.030 | 0.048 | 0.026 | 0.081 | 0.024 | 0.035 | 0.015 | 0.040 |
| | Ours (Sim(3), $w = 16$) | 0.112 | 0.045 | 0.123 | 0.261 | 0.022 | 0.137 | 0.044 | 0.044 | 0.016 | 0.089 |
| | Ours (Sim(3), $w = 32$) | 0.123 | 0.040 | 0.055 | 0.254 | 0.022 | 0.088 | 0.041 | 0.032 | 0.016 | 0.074 |
| | Ours (SL(4), $w = 8$) | 0.179 | 0.046 | 0.095 | 0.210 | 0.033 | 0.272 | 0.038 | 0.130 | 0.031 | 0.115 |
| | Ours (SL(4), $w = 16$) | 0.147 | 0.032 | 0.087 | 0.158 | 0.027 | 0.150 | 0.037 | 0.069 | 0.035 | 0.083 |
| | Ours (SL(4), $w = 32$) | 0.071 | 0.025 | 0.040 | 0.141 | 0.023 | 0.102 | 0.030 | 0.034 | 0.014 | 0.053 |

Table 7: Dense reconstruction evaluation on 7-Scenes [60] (unit: m).

| | Method | 7-Scenes | | | |
|---|---|---|---|---|---|
| | | ATE↓ | Acc.↓ | Complet.↓ | Chamfer↓ |
| Uncalib. | MASt3R-SLAM* [46] | 0.066 | 0.068 | 0.045 | 0.056 |
| | Ours (Sim(3), $w = 1$) | 0.070 | 0.066 | 0.051 | 0.059 |
| | Ours (Sim(3), $w = 8$) | 0.067 | 0.054 | 0.056 | 0.055 |
| | Ours (Sim(3), $w = 16$) | 0.067 | 0.054 | 0.058 | 0.056 |
| | Ours (Sim(3), $w = 32$) | 0.067 | 0.052 | 0.062 | 0.057 |
| | Ours (SL(4), $w = 1$) | 0.090 | 0.080 | 0.068 | 0.074 |
| | Ours (SL(4), $w = 8$) | 0.084 | 0.067 | 0.065 | 0.066 |
| | Ours (SL(4), $w = 16$) | 0.075 | 0.061 | 0.063 | 0.060 |
| | Ours (SL(4), $w = 32$) | 0.067 | 0.052 | 0.058 | 0.055 |

## B.2 Number of submaps per scene

As a reference, in Tables 8 and 9 we show the number of total submaps in each scene for 7-Scenes and TUM RGB-D for different values of experimented submap size, $w$, and also show the number of loop closures in each scene.

## B.3 Evaluation of Focal length Consistency

To provide quantitative results showing that VGGT can produce an estimate of the scene which differs by more than a similarity transformation to the true scene, in this section we show inconsistencies in estimates of camera intrinsics from VGGT. Here, a single camera is used per scene and different scenes can use different cameras. We observe that even though the true intrinsics of the camera should be approximately constant within a scene, VGGT has a varying estimate of the instrincs both inside a submap and across different submaps. This provides further demonstration that the VGGT reconstruction of a submap can differ from the true scene by more than a similarity transformation

Table 8: Window size $w$ and corresponding submap and loop closure counts when $w_{\mathrm{loop}} = 1$, shown as "# of submaps (# of loops)".

| Window size, $w$ | Sequences in 7-Scenes [65] | | | | | | |
|---|---|---|---|---|---|---|---|
| | chess | fire | heads | office | pumpkin | kitchen | stairs |
| 1 | 29 (11) | 50 (46) | 62 (49) | 58 (55) | 43 (37) | 43 (38) | 14 (12) |
| 8 | 4 (0) | 7 (4) | 8 (3) | 8 (4) | 6 (0) | 6 (2) | 2 (0) |
| 16 | 2 (0) | 4 (1) | 4 (2) | 4 (2) | 3 (0) | 3 (1) | 1 (0) |
| 32 | 1 (0) | 2 (0) | 2 (0) | 2 (0) | 2 (0) | 2 (0) | 1 (0) |

Table 9: Window size $w$ and corresponding submap and loop closure counts when $w_{\mathrm{loop}} = 1$, shown as "# of submaps (# of loops)".

| Window size, $w$ | Sequences in TUM-RGB-D [65] | | | | | | | | |
|---|---|---|---|---|---|---|---|---|---|
| | 360 | desk | desk2 | floor | plant | room | rpy | teddy | xyz |
| 1 | 168 (151) | 54 (42) | 98 (84) | 99 (87) | 102 (92) | 186 (162) | 95 (89) | 146 (125) | 56 (54) |
| 8 | 21 (4) | 7 (4) | 13 (7) | 13 (3) | 13 (5) | 24 (7) | 12 (10) | 19 (9) | 7 (5) |
| 16 | 11 (2) | 4 (2) | 7 (4) | 7 (2) | 7 (2) | 12 (4) | 6 (4) | 10 (4) | 4 (2) |
| 32 | 6 (1) | 2 (0) | 4 (2) | 4 (1) | 4 (2) | 6 (2) | 3 (1) | 5 (2) | 2 (0) |

and contain affine and projective degrees of freedom which can be resolved using the homography alignment. In Table 10 we summarize the standard deviation, range, and average of all focal length estimates for four scenes. We observe that for both the office loop scene and 7-Scenes, our $\mathrm{Sim}(3)$ variant of VGGT-SLAM performs comparable to the $\mathrm{SL}(4)$ variant while $\mathrm{SL}(4)$ performs significantly better than $\mathrm{Sim}(3)$ on the Tabletop and Bollards scene. Consistent with this observation, in Table 10, we notice that the later two have much larger intrinsic error (larger standard deviation and larger range) than the former two.

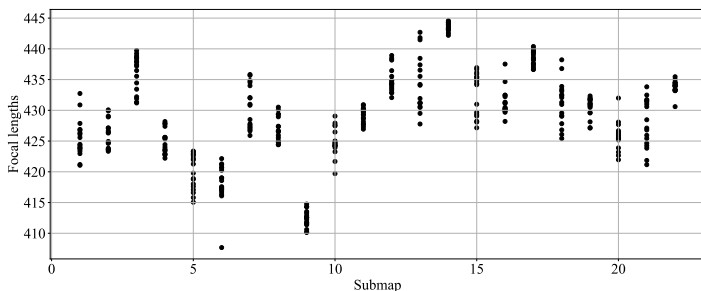

Figure 4: VGGT estimates of the focal length (fx) of every keyframe in the office loop scene from Fig. 2 for all 22 submaps.

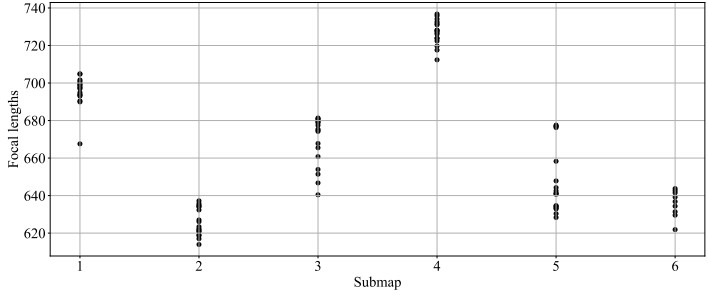

Figure 5: VGGT estimates of the focal length (fx) of every keyframe in the tabletop scene from Fig. 7 for all 6 submaps.

Table 10: Statistics of VGGT Focal length (fx) estimates. All values in pixels.

| Scene | Std Dev | Range | Average |
|---|---|---|---|
| Office Loop (Fig. 2) | 7.3 | 36.9 | 429.0 |
| 7-Scenes | 9.0 | 59.7 | 435.1 |
| Tabletop (Fig. 7) | 37.1 | 122.8 | 669.1 |
| Bollards (Fig. 8) | 51.8 | 177.3 | 738.9 |

# C   Extra Qualitative Results

## C.1   Extra examples of $\mathrm{SL}(4)$ versus $\mathrm{Sim}(3)$

While we have mentioned that the $\mathrm{Sim}(3)$ version of VGGT-SLAM often provides high quality reconstructions, here we provide additional examples of cases where $\mathrm{Sim}(3)$ is not sufficient and $\mathrm{SL}(4)$ is necessary to achieve consistent alignment across submaps.

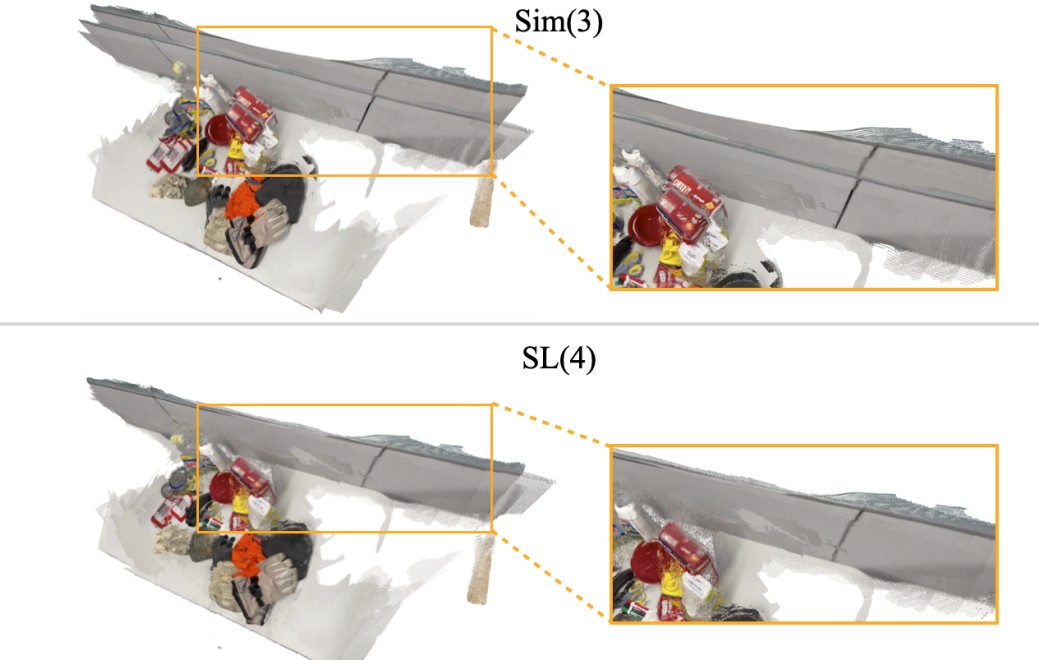

Figure 6: Example on a tabletop scene showing $\mathrm{Sim}(3)$ is unable to align the submaps while $\mathrm{SL}(4)$ is able to correct for projective ambiguity. Here $w = 32$ and $\tau_{\mathrm{disparity}} = 50$.

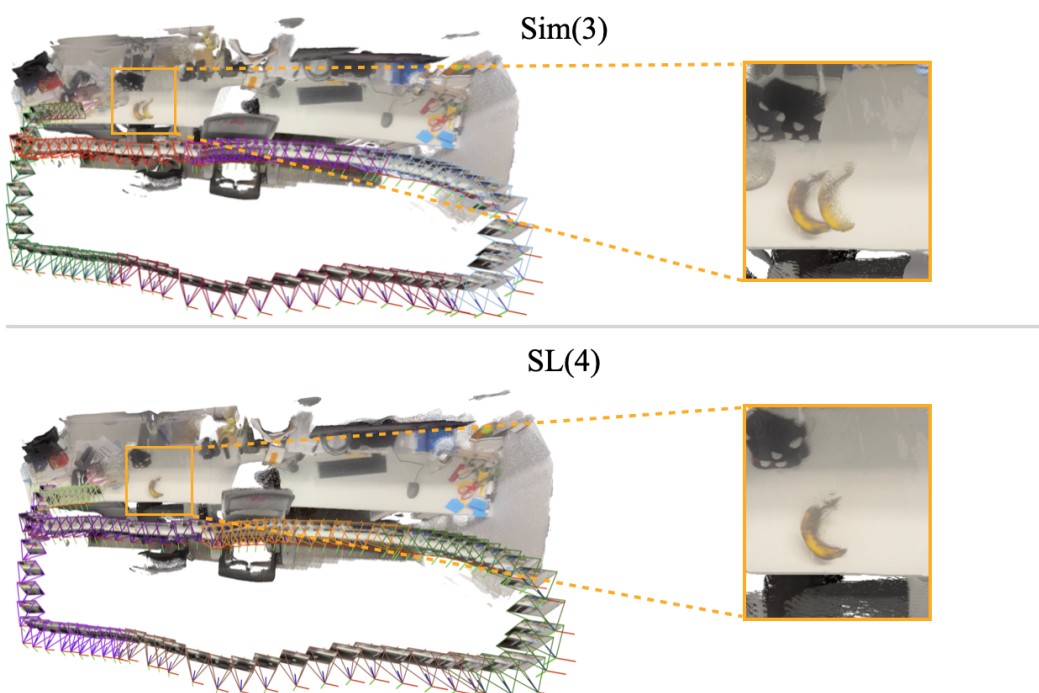

Figure 7: Example on a tabletop scene showing $\mathrm{Sim}(3)$ is unable to align the submaps while $\mathrm{SL}(4)$ is able to correct for projective ambiguity. The true scene only has one banana, but the $\mathrm{Sim}(3)$ reconstruction shows a hallucination of two caused by misalignment. Camera pose estimates are colored by submap. Here $w = 16$ and $\tau_{\mathrm{disparity}} = 50$.

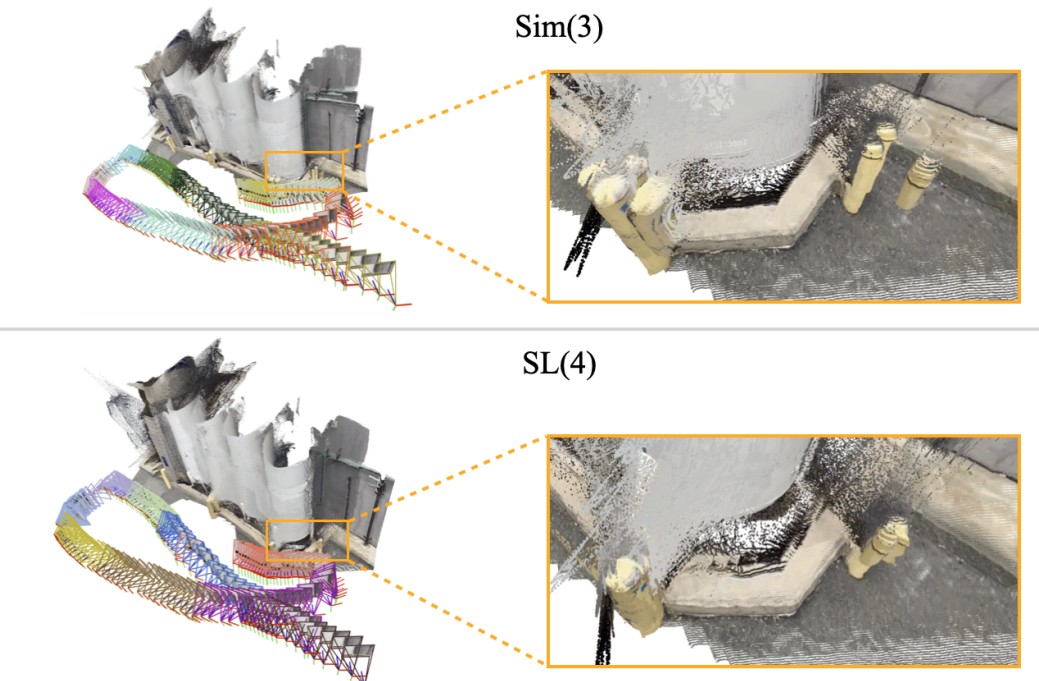

Figure 8: Example on an outdoor scene with yellow bollards surrounding tanks showing $\mathrm{Sim}(3)$ is unable to align the submaps while $\mathrm{SL}(4)$ is able to correct for projective ambiguity. The true scene has single bollards spaced around the tanks while the $\mathrm{Sim}(3)$ scene hallucinates clusters of bollards due to misalignment. Here $w = 16$ and $\tau_{\mathrm{disparity}} = 25$.

## C.2 7-Scenes Qualitative Results

Here we provide additional visualizations of scene reconstructions from the 7-Scenes dataset experiments for VGGT-SLAM with SL(4). We use the default parameters from Sec. 5.

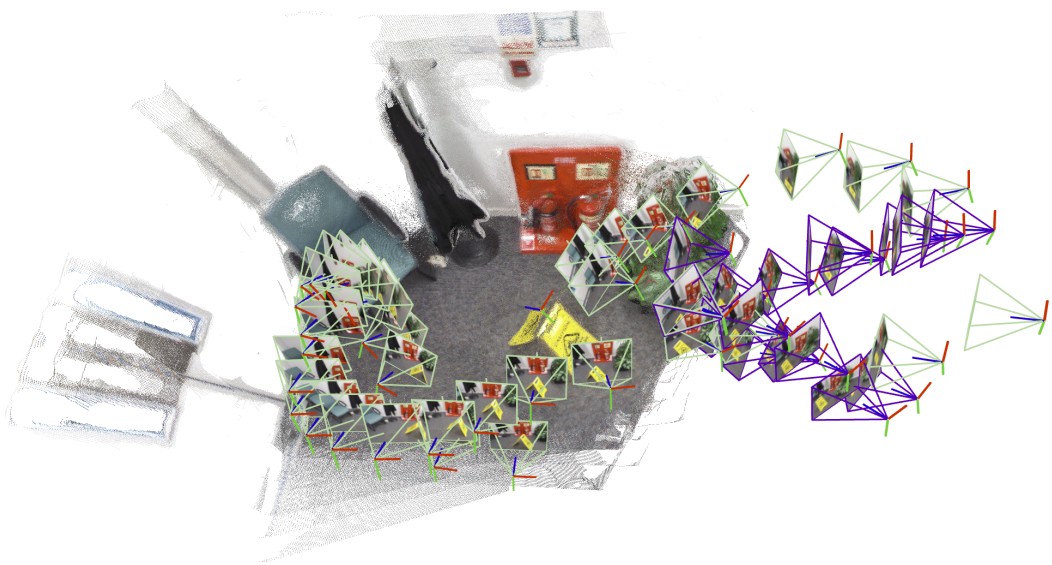

Figure 9: Visualization of reconstruction on 7-Scenes `fire` scene with 2 submaps. Camera pose estimates are colored by submap.

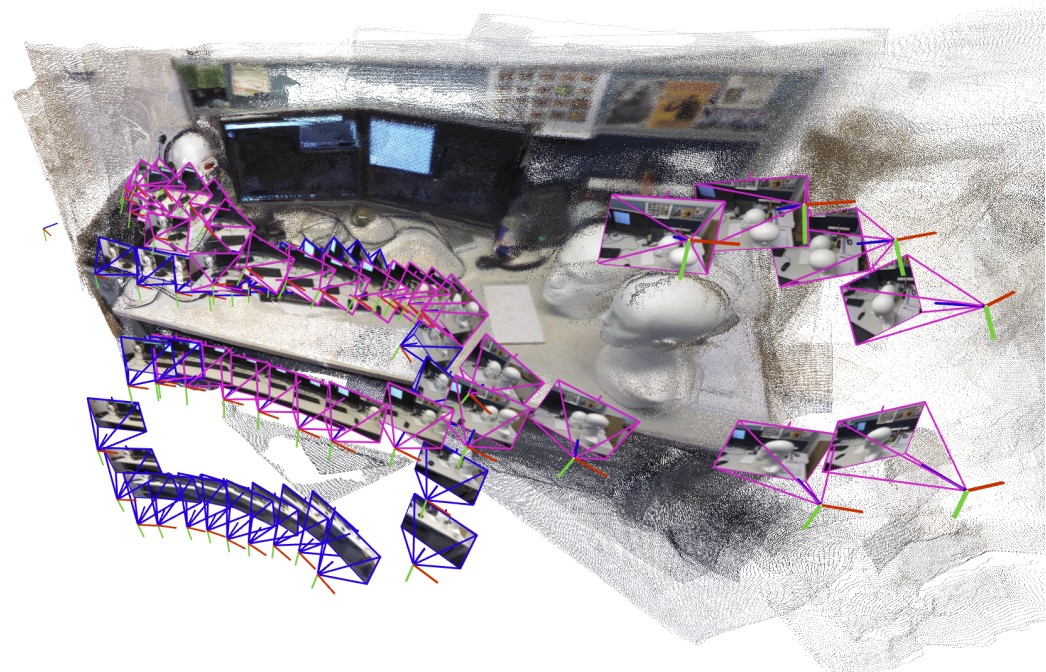

Figure 10: Visualization of reconstruction on 7-Scenes `heads` scene with 2 submaps. Camera pose estimates are colored by submap. Part of the scene in cropped for visual clarity.

## C.3 TUM RGB-D Qualitative Results

Here we provide additional visualizations of scene reconstructions from the TUM RGB-D dataset experiments for VGGT-SLAM with $\mathrm{SL}(4)$. We use the default parameters from Sec. 5.

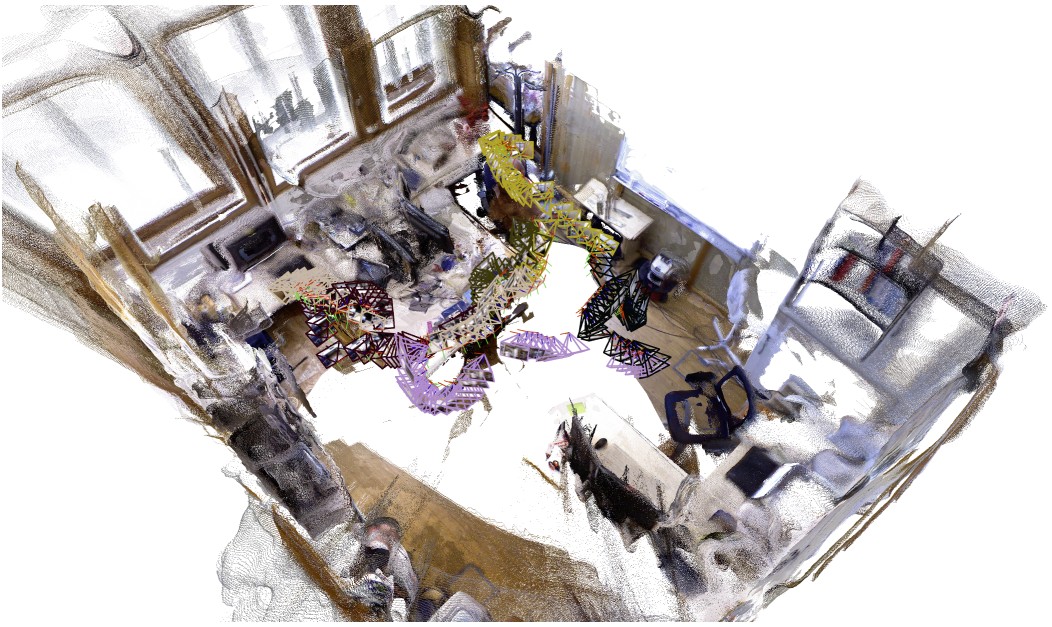

Figure 11: Visualization of reconstruction on TUM `room` scene with 6 submaps. Camera pose estimates are colored by submap. Part of the scene in cropped for visual clarity.

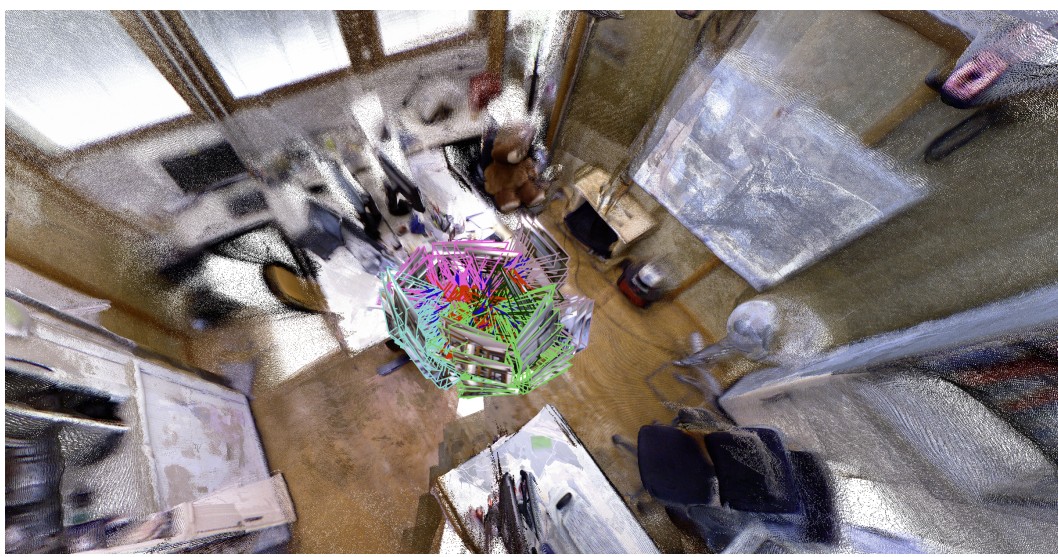

Figure 12: Visualization of reconstruction on TUM `360` scene with 6 submaps. Camera pose estimates are colored by submap. Part of the scene in cropped for visual clarity.

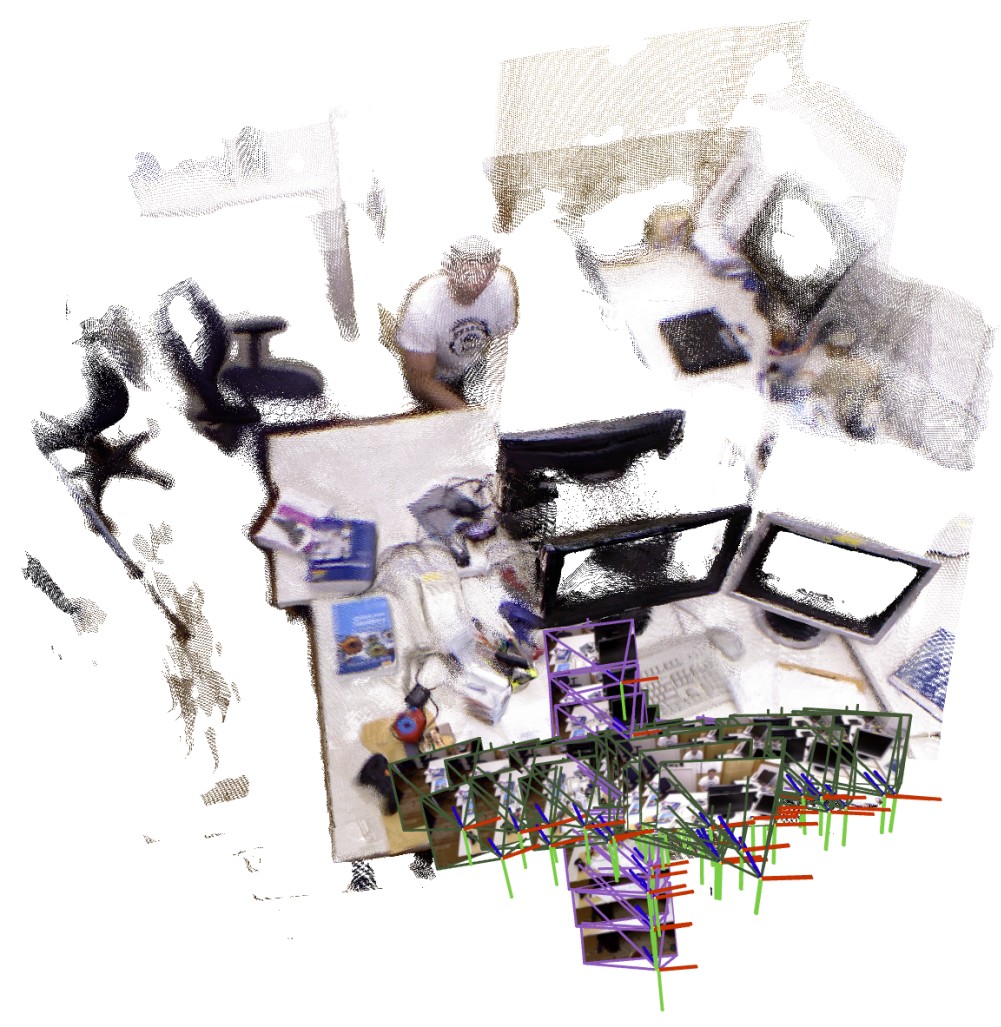

Figure 13: Visualization of reconstruction on TUM $\mathtt{xyz}$ scene with 2 submaps. Camera pose estimates are colored by submap.

## C.4 Additional Outdoor Qualitative Results

While our method is primarily tested on indoor scenes, here we provide an additional example of VGGT-SLAM on an outdoor scene from the TartanAir dataset [74]. Here $w = 16$, $\tau_{\text{disparity}} = 50$, and $\tau_{\text{conf}} = 50$.

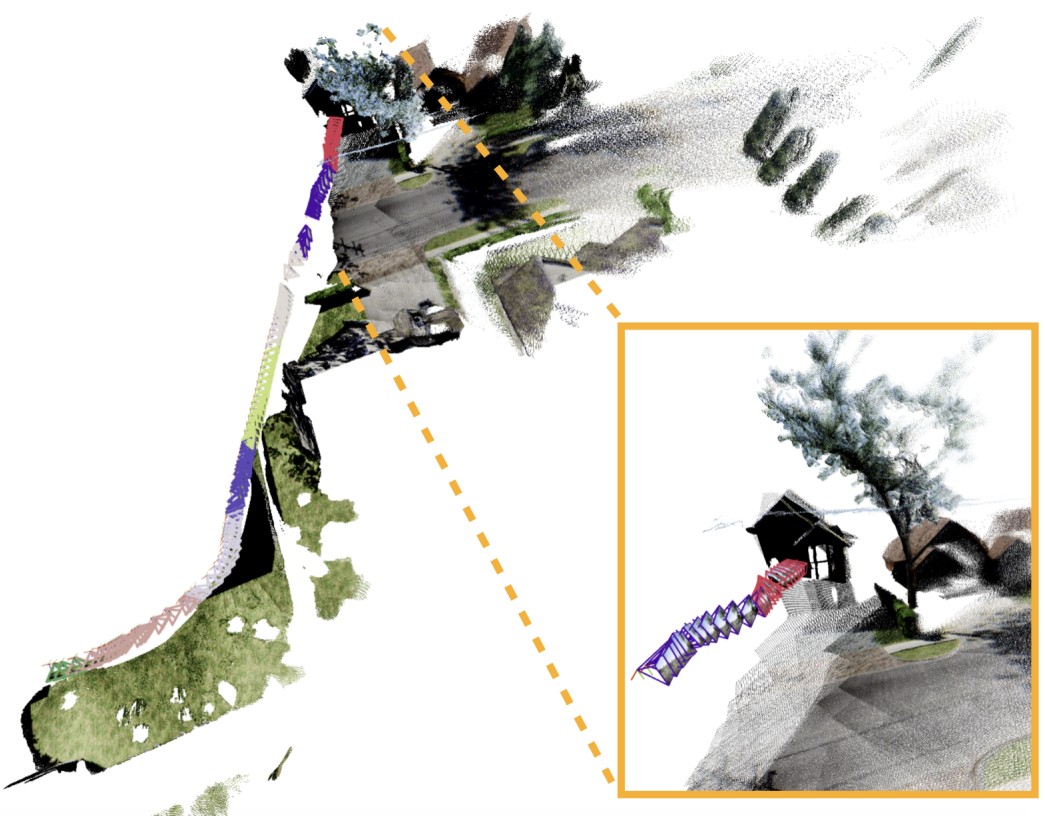

Figure 14: Visualization of reconstruction on TartanAir (scene *Neighborhood Easy, P005, left camera*) with 8 submaps. Camera pose estimates are colored by submap.

