# OpenReview forum: "VGGT-SLAM: Dense RGB SLAM Optimized on the SL(4) Manifold"
_NeurIPS.cc/2025/Conference — NeurIPS 2025 poster_

### Official Review · Reviewer_Wbeo · 2025-06-22

**Clarity:** 2
**Significance:** 3
**Originality:** 4
**Rating:** 4
**Confidence:** 4

**Summary:**

This paper proposes an approach to monocular SLAM by leveraging the output of VGGT, combined with a homography-based graph optimization strategy. The method aims to resolve projective ambiguities between submaps and improve global consistency.

**Questions:**

Section 4.2 (Transformation of camera poses via homography): A standard camera matrix is a 3×4 matrix, and if extended to a 4×4 form, its last row should be (0,0,0,1). After applying a general homography matrix, how do you ensure that this structure is preserved? Shouldn't the transformation between two camera matrices be expressed as Pi= Pj H^{-1}, where both Pi and Pj are 3×4 matrices?

**Ethical Concerns:**

["NO or VERY MINOR ethics concerns only"]

**Final Justification:**

The authors provide additional details on the runtime analysis and clarify their choice of using SL(4) instead of PGL(4). Although the proposed SL(4) optimization yields only marginal improvements for SLAM performance, the underlying concept of projective ambiguity is  quite interesting, offering valuable inspiration to the research community.

**Limitations:**

yes

**Quality:**

3

**Strengths And Weaknesses:**

**Strengths**

- This is the first work to use VGGT as a front-end for monocular SLAM, and also the first to utilize the Lie algebra of SL(4) for pose graph optimization.
- It introduces a new method for aligning submaps generated by feed-forward 3D models, addressing projective inconsistencies.
- The pose graph optimization is formulated using second-order Levenberg–Marquardt (LM) optimization over the Lie algebra of SL(4).
- The proposed method achieves better or competitive performance compared to current SOTA methods in terms of camera trajectory accuracy and reconstruction quality.


**Weaknesses**
- The paper assumes that the main reason of the global disalignment is the inaccurate intrinsics, but it lacks analysis on the quality of intrinsics from the output of VGGT
- The paper lacks an evaluation of runtime efficiency and GPU memory consumption, both of which are critical for SLAM applications.
- The 4×4 homography matrix used in the paper actually belongs to the projective general linear group PGL(4), not SL(4), although both groups have 15 degrees of freedom.
- There is no direct comparison between the proposed method and the original VGGT. While VGGT alone may not handle large-scale scenes, comparisons on smaller datasets would still provide valuable insights.
- Minor typo: Line 147 — “a single frames” should be “a single frame.”

---

> ### Author Rebuttal · Authors · 2025-07-31
>
> We thank the reviewer for providing detailed feedback and positive discussion of the paper. We address all of the reviewer comments in the following:
>
> [Analysis of intrinsics accuracy] We thank the reviewer for raising an important point about including analysis of the accuracy of intrinsic estimates from VGGT. In preparing VGGT-SLAM, we studied the estimated intrinsics from VGGT and noticed that for images taken from the same camera with constant calibration, focal length estimates from VGGT can have significant variation. We will include plots showing intrinsics estimates compared to the true intrinsic parameters in the revised manuscript for multiple scenes.  We summarize some of the results of VGGT’s estimate of the focal length (fx in pixels) of all keyframes in a scene, where a single camera is used for all images in a scene and thus should have constant focal length. 7-Scenes - avg: 435.1, std: 9.0, range: 59.7; Tabletop (Figure 5 in Supplementary Material) - avg: 679.4, std: 37.7, range: 117.3; Outdoor (Figure 6 in Supplementary Material) - avg: 738.9, std: 51.8, range: 177.3; Office Loop (Figure 2) - avg: 429.4, std: 6.9, range: 33.4. In particular, for the Tabletop and Outdoor scenes (for which in our paper we qualitatively demonstrate improvement with SL(4) over Sim(3)), there is substantial discrepancy in the focal length estimates from VGGT.
>
> [Analysis of runtime and memory] The reviewer raises an important point about including timing results for VGGT-SLAM. On an RTX 4090, VGGT-SLAM with SL(4) for submap size 16 takes 991.6 ms per submap with the following timing breakdown (all in milliseconds): VGGT: 820.2 ms, keyframe detection: 90.7 ms, loop closure detection and SALAD computation: 41.8 ms, relative homography estimation: 31.4 ms, and backend optimization: 7.4 ms. For Sim(3), all steps are the same except the relative estimation takes 11.3 ms instead of 31.4 ms. We will include a table showing a breakdown of timing results in the revised manuscript. Will will also include results showing GPU memory usage compared to submap size, of which the majority of memory is used by VGGT.
>
> [PGL(4) vs SL(4)] We thank the reviewer for detailed feedback on the usage of the PGL(4) group compared to the SL(4) group. There is an important but subtle distinction between the two which justify why we use SL(4) as the manifold for VGGT-SLAM optimization. The 4x4 homography matrix is only up to scale and the reviewer is correct that it belongs to PGL(4). However, the key point is that to optimize multiple homographies in the factor graph, we map each homography to a matrix in SL(4) and perform the optimization on the SL(4) manifold. This is because when computing the residual, it is necessary to compute the distance between homography matrices in a unique manner. For this reason, the matrices are scaled to have determinant one, which maps them to a unique SL(4) matrix, and the optimization occurs on the SL(4) manifold. This can be seen in prior works which perform optimization with the 3x3 homography using SL(3), such as [21] “Homography estimation on the special linear group based on direct point correspondence” - in particular Section II C discusses associating homography matrices to SL(3) . Our manuscript includes a larger set of references in this line of work in the last paragraph of Sec. 2 as follows: “Prior works use optimization on the SL(3) manifold (corresponding to the 8-DOF homography matrix commonly used in image alignment) for aligning multiple images for panoramic stitching [21, 59, 41, 40, 42, 34] and dense SLAM [33].” To increase clarity about this important point, we will change our discussion in the revised manuscript to state that the 4x4 homography matrix belongs to PGL(4) and that we associate the homography matrix to a unique matrix on SL(4) by normalizing it to have determinant one. We believe these revisions will increase the precision of the paper’s description of the homography.
>
> [Direct comparison to VGGT] To fully address the concern raised by the reviewer about not directly comparing to VGGT, we will add comparison results of VGGT compared to VGGT-SLAM. As the reviewer mentions, this will not be possible on larger scenes due to VGGT memory constraints, but we can include results on smaller scenes. Specifically, using the same images selected as keyframes by VGGT-SLAM to downsample the number of frames, we will include comparison results on 7-Scenes for VGGT. Doing this strategy on the TUM RGB-D dataset would exceed memory limits.
>
> [Typo line 147] We thank the reviewer for bringing this to our attention and will fix the typo in the revised manuscript.
>
> [Transformation of camera poses via homography] The reviewer is correct that there is a mistake in the original manuscript and $P_i$ and $P_j$ should be 3x4 matrices where $P_i= P_j H^{-1}$. Our attached code in the Supplementary Material uses the correct equation and we will fix the mistake in the paper. We thank the reviewer for pointing out this important fix.

---

> > ### Comment · Reviewer_Wbeo · 2025-08-03
> >
> > Thanks for the detailed reply. I have another question: according to the supplementary, Sim(3) pose graph with submap size 8 achieve same or better results than the setting SL(4) with submap size 32 for both TUM-RGBD and 7scenes (tab.4, tab.5, tab.6), then is this SL(4) alignment really necessary? Or is there any direct comparasion of these two particular settings that can prove the SL(4) is better?

---

> > > ### Author Response · Authors · 2025-08-03
> > >
> > > We provide two version of VGGT-SLAM - Sim(3) and SL(4) since Sim(3) is sometimes sufficient. For example, as you mentioned, 7-Scenes and TUM RGB-D shows comparable performance between our Sim(3) and SL(4) versions. There are two observations that we emphasis in our manuscript: the first is that despite the added degrees of freedom with SL(4), even on scenes where Sim(3) is sufficient, SL(4) can still perform comparable and sometimes slightly better. The second, and most important, is that there are scenes where VGGT has a projective ambiguity and SL(4) is required for VGGT-SLAM. For this, we show examples between VGGT-SLAM with Sim(3) and SL(4) in Figures 1, 4, 5, and 6.
> > >
> > > The analysis of intrinsic accuracy discussed in our first comment [Analysis of intrinsics accuracy], which we will add in our revised manuscript, helps to provide some additional context for the projective ambiguity as some scenes have very high VGGT errors in focal length estimates (std of 51.8 with range 177.3 in the case of Figure 6) while 7-Scenes has less (std of 9.0 with range 59.7)
> > >
> > > In summary, if VGGT produces scenes without the projective ambiguity, then a Sim(3) transformation is sufficient and SL(4) can also be used since Sim(3) is a subset of SL(4). However, we need SL(4) for VGGT-SLAM in general because as we demonstrate using the Projective Reconstruction Theorem, there can be projective ambiguity from VGGT since the camera calibration is not known to the model, and Sim(3) is insufficient (such as in the above mentioned figures).

---

> > > > ### Comment · Area_Chair_RVse · 2025-08-08
> > > >
> > > > Dear Reviewer Wbeo,
> > > >
> > > > Today is the last day to engage in a discussion with the authors and they have replied to your recent concerns. Please engage as soon as possible.
> > > >
> > > > Best,
> > > >
> > > > AC

---

### Official Review · Reviewer_BmpF · 2025-06-25

**Clarity:** 3
**Significance:** 3
**Originality:** 3
**Rating:** 5
**Confidence:** 4

**Summary:**

This paper introduces VGGT-SLAM, an innovative dense RGB SLAM system that cleverly leverages VGGT's feed-forward scene reconstruction capabilities to construct dense maps from uncalibrated monocular cameras incrementally. The author proposes the first factor graph formulation that operates directly on the SL(4) manifold to address projective ambiguity. Even in practical scenarios, where projective ambiguity is less dominant, 62 we show that SL(4)-based optimization achieves performance competitive with or superior to other 63 state-of-the-art learning-based SLAM approaches, offering a principled framework for handling cases where similarity transformations are insufficient.

**Questions:**

1. Based on Table 3 showing Root Mean Square Error (RMSE) reconstruction evaluation on 7-Scenes, the performance differences between Sim(3) and SL(4) optimization appear to be minimal. This raises a fundamental question about the necessity of the complex SL(4) manifold optimization. Can the authors provide a comprehensive ablation study comparing:
   - SE(3) pose graph optimization
   - Sim(3) similarity transformation
   - Full SL(4) homography optimization
This would help clarify whether the SL(4) formulation is truly necessary or if simpler approaches could achieve comparable results.
2. What is the computational cost difference between these approaches, and does the marginal accuracy improvement justify the increased complexity? The similar performance between Sim(3) and SL(4) on 7-Scenes raises suspicions about potential training data contamination. Can the authors explicitly confirm whether the 7-Scenes data was used in VGGT's training? If so, what percentage of the training data comes from 7-Scenes?
3. How does VGGT-SLAM perform on scenes that are fundamentally different from the training distribution (e.g., outdoor vs. indoor, different lighting conditions, different camera intrinsics)?
4. How does the computational efficiency of VGGT-SLAM compare to Traditional SLAM methods (ORB-SLAM3, DROID-SLAM)? Can the author provide a detailed timing breakdown that shows reasonable computational costs? This clarification is crucial for understanding the true contribution and practical value of the proposed approach.

**Ethical Concerns:**

["NO or VERY MINOR ethics concerns only"]

**Final Justification:**

While the authors' responses have addressed most of my concerns, I raise my original rating to AC.

**Limitations:**

Yes

**Quality:**

3

**Strengths And Weaknesses:**

## Strengths
- **Large-Scale Scene Extension**: VGGT was originally limited by GPU memory constraints to processing only a small number of frames. The authors successfully extended it to large-scale scenes by creating multiple submaps and performing global alignment. This breakthrough enables the reconstruction of scenes requiring hundreds or even thousands of frames.

- **SL(4) Manifold Optimization**: To address the projective ambiguity issue in uncalibrated camera reconstructions, the paper proposes optimization over the SL(4) manifold. This approach estimates 15-degree-of-freedom homography transforms between consecutive submaps, not only accounting for potential loop closure constraints but also providing more accurate alignment compared to traditional similarity transformations (Sim(3)).

- **Extensive experiments** on standard RGB SLAM benchmarks demonstrate the effectiveness of VGGT-SLAM. The system exhibits robust performance across various scenarios, including long video sequences that are infeasible for VGGT due to its high GPU requirements.

## Weaknesses

- **Degenerate Cases**: The paper acknowledges that the estimation of the full 15-DOF homography matrix can be degenerate in the case of planar points, leading to unstable solutions. This is observed in the planar floor scene of the TUM dataset, where the performance of VGGT-SLAM degrades due to the presence of degenerate cases.
- **Outlier Sensitivity**: The current implementation of homography using points from VGGT is vulnerable to outliers. While the use of a 5-point RANSAC helps reduce this issue, high outlier ratios or adversarial outliers can still cause incorrect homography estimates, affecting the overall performance of the system.
- **Computational Efficiency**: The paper does not extensively discuss the computational efficiency of the proposed method. While the experiments are performed on a high-end GPU, the scalability and real-time performance of VGGT-SLAM in resource-constrained environments are not fully explored.

- **Complexity of Nonlinear Optimization**: The paper introduces nonlinear factor graph optimization on the SL(4) manifold to compute absolute homographies and align submaps globally. While this approach is mathematically rigorous and provides a principled framework for handling projective ambiguity, it introduces additional complexity and potential robustness issues. Specifically, the iterative computation of state increments through solving linearized least squares problems can be computationally intensive and sensitive to initialization and noise. Integrating a nonlinear optimization system with the end-to-end VGGT model may compromise the overall simplicity and elegance of the approach. The additional optimization steps could complicate system implementation and deployment, potentially reducing accessibility for practitioners. The authors might consider exploring more streamlined and elegant ways to combine VGGT's strengths with global alignment, without relying on complex nonlinear optimization frameworks.

---

> ### Author Rebuttal · Authors · 2025-07-31
>
> We thank the reviewer for positive feedback and for insightful questions which we address in the following:
>
> [Degenerate Cases and Outlier Sensitivity] The reviewer mentions two important areas of improvement for VGGT-SLAM, which are discussed in detail in Sec. 6 Limitations. While these issues exist, we are still able to show state-of-the-art results on a variety of datasets. For example, even though we have lower accuracy on the floor scene from TUM RGB-D due to the planarity of the scene, our average across all TUM RGB-D scenes is still the most accurate compared to all uncalibrated baselines. Solving the planar degeneracy issue would further boost performance and reduce potential failure cases. In practice, our 5-point RANSAC procedure with estimating homography is shown to be effective throughout our experiments, but we see potential to use inspiration from ideas such as ray-based matching from MASt3R-SLAM to further add robustness to outliers. We believe these each are interesting and non-trivial areas and we plan to explore them in future work to further improve performance.
>
> [Computational Efficiency] The reviewer raises an important point about including timing results for VGGT-SLAM. On an RTX 4090, VGGT-SLAM with SL(4) for submap size 16 takes 991.6 ms per submap with the following timing breakdown (all in milliseconds).  VGGT: 820.2 ms, keyframe detection: 90.7 ms, loop closure detection and SALAD computation: 41.8 ms, relative homography estimation: 31.4 ms, and backend optimization: 7.4 ms. For Sim(3), all steps are the same except the relative estimation takes 11.3 ms instead of 31.4 ms. We will include a table showing a breakdown of timing results in the revised manuscript. We will also include results showing GPU memory usage compared to submap size, of which the majority of memory is used by VGGT. The overall framerate of VGGT-SLAM is also comparable to MASt3R-SLAM on the same hardware. The majority of VGGT-SLAM’s compute requirements are for VGGT and given the popularity around VGGT, it can be expected that there will be future lighter weight version of VGGT that can be plugged into VGGT-SLAM; although, running on lightweight devices such as edge compute is beyond the scope of the current paper.
>
> [Complexity of nonlinear optimization] We thank the reviewer for raising this important concern about the computational complexity of the nonlinear backend optimization. The computation time of the nonlinear solver (built on top of GTSAM) is about 7.4 ms per submap which uses less than 1 percent of the total submap computation time. Typically, real time mapping systems with GTSAM have on the order of 10s of thousands of factors for bearing measurements while we just have factors between adjacent submaps and submaps with loop closures. As discussed, to clarify this important point, we will include a timing table which shows the time used by each major component of VGGT-SLAM. To minimize the amount of user complexity, we chose to integrate our backend optimization as part of a popular optimization library (GTSAM) which creates a simpler python API interface between the core VGGT-SLAM code and the backend optimization steps. Future work could potentially leverage foresight of projective ambiguity discussed in VGGT-SLAM as a non-trivial but exciting direction to improve training, and as the reviewer mentions, this can further reduce the number of components of the SLAM system.
>
> [Table 3 Results] Since Sim(3) optimization is sometimes sufficient with submap alignment, we include both the Sim(3) and SL(4) versions of VGGT-SLAM. While results in Table 3 are comparable between Sim(3) and SL(4), we include examples of cases where Sim(3) is not sufficient due to projective ambiguity. To reconcile this limitation of Sim(3), we develop the full SL(4) version of VGGT-SLAM while concurrently using theory from classical computer vision to justify why there can be a projective ambiguity in the submap alignment problem. Our manuscript provides two examples in Figure 1 and we include three more examples in the Supplementary Material. For an ablation study of Sim(3) compared to SL(4), we show both pose evaluation and dense reconstruction evaluations of both the Sim(3) and SL(4) variants of VGGT-SLAM on 7-Scenes and TUM RGB-D datasets. Furthermore, we include additional quantitative results on 7-Scenes and TUM RGB-D in the Supplementary Material for varying submap sizes, along with qualitative examples. An SE(3) baseline would not be expected to show reasonable results since VGGT does not try to estimate the scale of the scene and thus scale must be estimated between submaps to get a reasonable overall map.
>
> To include more detailed analysis,  in our revised paper we will also include an additional plot showing the fluctuations in VGGT’s estimate of the camera intrinsics. The fluctuation can be attributed to the projective ambiguity.  We summarize some of the results of VGGT’s estimate of the focal length (fx in pixels) of all keyframes in a scene, where a single camera is used for all images in a scene and thus should have constant focal length. 7-Scenes - avg: 435.1, std: 9.0, range: 59.7; Tabletop (Figure 5 in Supplementary Material) - avg: 679.4, std: 37.7, range: 117.3; Outdoor (Figure 6 in Supplementary Material) - avg: 738.9, std: 51.8, range: 177.3; Office Loop (Figure 2) - avg: 429.4, std: 6.9, range: 33.4. In particular, for the Tabletop and Outdoor scenes (for which in our paper we qualitatively demonstrate improvement with SL(4) over Sim(3)), there is substantial discrepancy in the focal length estimates from VGGT.
>
> [Computational Cost Difference] We will clarify this important question about the computation cost difference between SL(4) and Sim(3) by including the timing breakdown table discussed earlier. Specifically, for Sim(3) optimization, all steps of VGGT-SLAM are the same except the relative estimation takes 11.3 ms instead of 31.4 ms, leading to approximately 20.1 ms of increased cost per submap for SL(4) with our current implementation of the VGGT-SLAM code.
>
> [7-Scenes dataset] Based on the VGGT paper, the following datasets were included in training and are similar to those used by MASt3R (which is the Geometric Foundation Model used by one of the compared methods, MASt3R-SLAM). The datasets used in training are: Co3Dv2, BlendMVS, DL3DV, MegaDepth, Kubric, WildRGB, ScanNet, Hyper- Sim, Mapillary, Habitat, Replica, MVS-Synth, PointOdyssey, Virtual KITTI, Aria Synthetic Environments, Aria Digital Twin. The 7-Scenes dataset is not included in the VGGT training. We also include visual results on custom captured scenes such as the office loop (Fig 2) and several cell phone captured scenes in both indoor and outdoor in Appendix C. The observed similarity between Sim(3) and SL(4) performance is because Sim(3) is a subset of the SL(4) group and if a submap alignment can be fully explained by a Sim(3) transformation, then the SL(4) transformation should be nearly identical to the Sim(3) transformation.
>
> [Using VGGT-SLAM on out-of-distribution data] We use the frozen weights from VGGT and experiment on both popular benchmarking datasets and custom datasets which have been collected by multiple cameras such as a cell phone camera and a RealSense D455 camera. Our custom datasets include both indoor scenes and an outdoor scene (Figure 6 in the Supplementary Material). In developing VGGT-SLAM, we experimented with other scene challenges such as an outdoor scene at night being illuminated by flood lights and in general had good results. However, while VGGT-SLAM can correct projective ambiguity from VGGT, which may be in part due to challenging scenes, avoiding an otherwise completely incorrect scene reconstruction in an out-of-distribution scene would be a credit to VGGT not VGGT-SLAM. In practice, we observe impressive generalizability of VGGT, which has made it a powerful foundation model for a SLAM system.
>
> [Comparison to Traditional SLAM methods] Our manuscript includes comparisons to DROID-SLAM, ORB-SLAM, and other traditional SLAM approaches in Table 2 in terms of accuracy. For timing, we will include a timing breakdown of VGGT-SLAM as discussed. For reference, the timing of VGGT-SLAM per frame is comparable to that of MASt3R-SLAM and we will include timing results of MASt3R-SLAM as well in the revised manuscript. ORB-SLAM is much lighter in computation than all of VGGT-SLAM, MASt3R-SLAM, and DROID-SLAM. However, these works are trying to accomplish different ends by also using different means. ORB-SLAM is designed to be a lightweight CPU-only visual odometry system that requires calibrated cameras and many complex components. In contrast, methods like VGGT-SLAM and MASt3R-SLAM are dense SLAM systems which can work with uncalibrated cameras and remove multiple degrees of complexity by extending recent Geometric Foundation Models to a SLAM system. Due to recent popularity around Geometric Foundation Models such as VGGT, it is likely that lighter weight versions will soon be available, and since we do not require fine tuning, we look forward to easily integrating future more efficient VGGT models into VGGT-SLAM.

---

> > ### Comment · Reviewer_BmpF · 2025-08-04
> >
> > The authors adequately addressed most concerns, particularly with promised additional experiments and timing analyses

---

> ### Comment · Area_Chair_RVse · 2025-08-04
>
> Could you submit your mandatory acknowledgment? Thanks!
>
> AC

---

### Official Review · Reviewer_wLsM · 2025-06-30

**Clarity:** 3
**Significance:** 2
**Originality:** 3
**Rating:** 5
**Confidence:** 5

**Summary:**

This paper addresses the alignment of local reconstructions and camera poses produced by feed-forward transformer models—specifically VGGT—into a globally consistent map and trajectory. Since VGGT reconstructs each submap from uncalibrated video, these local reconstructions can suffer from projective distortions that make simple similarity (Sim(3)) alignment inaccurate. To overcome this, the authors propose aligning every pair of local submaps with a 4×4 homography on the SL(4) manifold. Relative homographies are estimated both between temporally adjacent submaps and between submaps linked by loop-closure detection. These pairwise homographies form the edges of an SL(4) pose graph, whose global optimization yields absolute 4×4 transformations. Finally, the global camera poses are recovered by decomposing these absolute homographies into intrinsic and extrinsic parameters. Experiments on the 7-Scenes and TUM-RGBD datasets—evaluating both global trajectory accuracy and dense point-cloud alignment—show that this SL(4)-based approach outperforms previous methods when camera calibration is unavailable.

**Questions:**

- Please refer to weakness.
- Eq.2: The paper only describe a minimal solution with RANSAC, is there any non-linear refinement after that? If not, what is the performance difference between w/ and w/o non-linear refinement.
- What is the efficiency / fps of the full pipeline?
- To distinguish between the insufficiency of SIM(3) transformation itself and the estimation of SIM(3), what if we initialize SIM(3) pose (both relative and global) with decomposed Homography and then refine?

**Ethical Concerns:**

["NO or VERY MINOR ethics concerns only"]

**Final Justification:**

This paper proposes to aligns projectively distorted point clouds from VGGT both locally and globally using 4\times4 homography transformation. The motivation is to address the camera intrinsic ambiguity of VGGT's prediction, and the corresponding solution is simple and proven to be effective on limited small scale scenes. During the rebuttal, the authors have indicated that they have verified the advantage of SL(4) homography transformation over SIM(3) similarity transformation is still valid when the distortion is more than projective, where the conclusion is independent of specific estimation method of SIM(3).
In addition, the pipeline can be made differentiable with reasonable effort. Therefore, I recommend to accept the paper.

**Limitations:**

The authors have discussed the limitations of planar scenes. However, the author did not discuss the potential limitation when the distortion of the VGGT local reconstruction is more than a projective transformation.

**Paper Formatting Concerns:**

I do not have formatting concern.

**Quality:**

3

**Strengths And Weaknesses:**

### Strengths
- This paper is well organized and written. The problem of projective ambiguity in VGGT is clear and motivates the use of a $4\times4$ homography transformation.

- The method itself is simple, effective, and presented with sufficient detail. The open-sourced code further enhances reproducibility. The implementation is also comprehensive, including relative homography estimation using DLT and global homography estimation as a pose graph.

- The proposed solution is agnostic to local 3D reconstruction methods and can be combined with other uncalibrated 3D reconstruction models.

### Weaknesses
- The major assumption motivating the paper is that when Sim(3) is insufficient, the relative transformation between local VGGT reconstructions is primarily a projective transformation due to uncalibrated cameras. However, this assumption is insufficiently justified in the current version of the paper. Specifically, the following questions merit further clarification:

  - How frequently are VGGT’s reconstructions distorted?

  - When a reconstruction is distorted, how well can a projective transformation handle this distortion?

  - How reliable is the decomposition itself—i.e., how the accuracies of decomposed K, R, and t related to the accuracy of H and the magnitude of shearing, stretching, and perspective components in H?

- The method is non-learnable. To engage a broader audience at NeurIPS, it would be worthwhile to discuss and better demonstrate how the proposed method could be integrated into a learning pipeline. Both the estimation of local and global homographies could be made differentiable with reasonable effort.

- A discussion of classic projective reconstruction (e.g., Factorization Methods for Projective Structure and Motion, CVPR ’96; Projective Bundle Adjustment from Arbitrary Initialization using the Variable Projection Method, ECCV ’16) is missing.

- The datasets used in evaluation are limited, only TUM-RGBD and 7 Scenes. While other standard benchmarks such as Tartan Air and KITTI is missing. Therefore, it is uncertain whether the assumption still holds for outdoor scenes with forward motion.

---

> ### Author Rebuttal · Authors · 2025-07-31
>
> We thank the reviewer for an accurate summary and positive comments, and we address all questions and concerns as follows:
>
> [Frequency that VGGT’s reconstructions are distorted] In addition to the two visual examples in Figure 1 of scenes where Sim(3) is insufficient for submap alignment and SL(4) is needed, we also include three additional examples in the Supplementary Material. Since Sim(3) is sometimes sufficient, we include our simpler Sim(3) version of VGGT-SLAM as well in the paper and in our attached code (where a single command line option allows switching to Sim(3)). However, because we notice multiple important cases where Sim(3) is not sufficient, we show that a higher dimensional projective alignment is needed and hence develop the full SL(4) version of VGGT-SLAM.
>
> While it can be difficult to precisely quantify how often SL(4) is needed, to better clarify the benefits of SL(4), in our revised paper we will include a plot showing the fluctuations in VGGT’s estimate of the camera intrinsics. The fluctuation can be attributed to the projective ambiguity.  We summarize some of the results of VGGT’s estimate of the focal length (fx in pixels) of all keyframes in a scene, where a single camera is used for all images in a scene and thus should have constant focal length. 7-Scenes - avg: 435.1, std: 9.0, range: 59.7; Tabletop (Figure 5 in Supplementary Material) - avg: 679.4, std: 37.7, range: 117.3; Outdoor (Figure 6 in Supplementary Material) - avg: 738.9, std: 51.8, range: 177.3; Office Loop (Figure 2) - avg: 429.4, std: 6.9, range: 33.4. In particular, for the Tabletop and Outdoor scenes (for which in our paper we qualitatively demonstrate improvement with SL(4) over Sim(3)), there is substantial discrepancy in the focal length estimates from VGGT.
>
> [How well can a projective transformation handle the distortion] We thank the reviewer for the insightful question which lets us clarify an important component of the theory behind VGGT-SLAM. According the Projective Reconstruction Theorem (section 4.2 of the manuscript and in [25]), if a set of uncalibrated cameras satisfies multi-view constraints (which can be given by the Fundamental matrix for corresponding points), then the difference between the estimated reconstruction of the scene and the true reconstruction can be explained by a 15 DOF projective transformation. Thus, a projective transformation is sufficient and in addition to theoretical justification, we experimentally observe that the SL(4) method is able to align submaps in the cases where Sim(3) is insufficient. If the reviewer is referring to camera lens distortion, we do not correct for camera distortion and distortion would violate the premise of the Projective Reconstruction Theorem since straight lines would not be preserved. VGGT also assumes negligible lens distortion.
>
> [Decomposition of K,R, and t] To clarify, we do not estimate the poses from scratch. Instead, we get the pose estimates from VGGT and apply a transforming homography to the poses. For this, we use the procedure outlined in the last section of 4.2.  However, we apologize that there was a typo in the equation in this section. The correct procedure to transform a camera matrix $P_j$ to $P_i$ should be $P_i=P_j H^{-1}$ where $P_i$ and $P_j$ are 3x4 matrices. Our code has the correct method but the typo in the paper likely caused confusion that a 4x4 matrix had to be decomposed and we apologize for that confusion. We have fixed the error in the revised manuscript. The decomposition of the matrix $P_i$ is then simply (we use decomposeProjectionMatrix in OpenCV) extracting the translation component, an SO(3) component, and an upper triangular component, for which we have not observed instabilities in practice.
>
> [Method is non-learnable] In this work, we extend VGGT to a SLAM system and in doing so we leverage theory from classical multi-view reconstruction of a scene with uncalibrated images to show why aligning submaps from VGGT cannot always be done effectively with a similarity transform and instead needs a projective transform. It would be straightforward to use an existing AutoDiff library for SLAM like PyPose or Theseus for the linear relative homography estimation and nonlinear global optimization, allowing these steps to either be learned or form a bilevel optimization problem to also support VGGT training. However, since it would be straightforward to do so, we do not believe it would add to the theoretical contribution of the work and would be out of scope. Additionally, with the popularity of geometric foundation models and of VGGT, many faster and more capable versions of VGGT are likely to become available and these can easily be integrated into VGGT-SLAM since no fine tuning is required with our current approach.
>
> [Missing references] We thank the reviewer for mentioning relevant prior works and we will add a discussion of these into our revised manuscript.
>
> [Outdoor Scenes] The reviewer mentions a concern that assumptions may not hold in the case of outdoor scenes or scenes with forward motion. Our response is five-fold. First, if the assumptions are referring to whether submaps can always be aligned with a projective transformation, we refer to our earlier remark that the Projective Reconstruction Theorem does not place requirements on the type of motion and thus the alignment procedure still works in the case of forward motion. Second, as an experiment of forward motion, we include results in a loop around an office floor in Figure 2 where multiple submaps are aligned under forward motion. Third, for outdoor scenes, our Supplementary Material in Figure 6 includes results of running VGGT-SLAM outside a building viewing a curbside with bollards and large holding tanks. Fourth, as discussed in the limitations section, we do not expect VGGT-SLAM with SL(4) to work well on long driving datasets like Kitti because as discussed in the limitation section, SL(4) can have excessive drift if there is a long period between loop closures as is the case in driving datasets. Lastly, in response to the reviewer, we have run a scene from TartanAir with VGGT-SLAM to provide an additional outdoor example and will include the results in the revised Supplementary Material.
>
> [Non-linear refinement] To clarify, in case there is confusion around our use of the word “minimal” to describe our RANSAC solver, we use a minimum solver for optimizing a local estimate of homography, meaning that we use the minimum number of points required for a unique solution (which is 5 in the case of a 4x4 homography). This can be solved with a linear solver and we employ a 5-point RANSAC to add robustness to outliers. Using a minimum number of points is well-known to be the most efficient way to use RANSAC since the fewer points required to obtain a solution, the less number of iterations needed to statistically reach a solution built with a set of inliers. After getting a solution from RANSAC, we attempted an additional optimization over all inliers, which is a common practice, but we did not find an accuracy improvement and thus removed the extra computation step. Our global optimization in the backend to optimize the factor graph does use a non-linear solver.
>
> [FPS of the full pipeline] We thank the reviewer for raising an important issue that we should have included timing results in the paper. We have added a full timing breakdown table to the revised manuscript which includes the following results: On an RTX 4090, VGGT-SLAM with SL(4) for submap size 16 takes 991.6 ms per submap with the following timing breakdown (all in milliseconds). VGGT 820.2 ms, keyframe detection: 90.7 ms, loop closure detection and SALAD computation: 41.8 ms, relative homography estimation: 31.4 ms, and backend optimization: 7.4 ms. For Sim(3), all steps are the same except the relative estimation takes 11.3 ms instead of 31.4 ms. Thus, VGGT-SLAM with SL(4) runs at approximately 16 FPS.
>
> [Distinguishing between Sim(3) error and Sim(3) insufficiency] We are not sure if we understand the question here and perhaps this is due to our typo of correcting the pose estimates with $P_i=P_j H^{-1}$ as discussed earlier.  If a Sim(3) (7DOF) transformation is not sufficient to align submaps, then either an affine (12 DOF) or a projective (15 DOF) transformation is needed. To start with Sim(3) and then refine the transformation in a way that could align the submaps would imply upgrading it to a higher dimensional transform (affine or projective). Our current formulation of Sim(3) estimation minimizes the potential for a false estimate of Sim(3) because we are aligning the last frame of one submap with the first frame of the subsequent submap, where the two frames are identical. In other words, the two frames have the same pose and we can therefore just set the rotation and translation of the first frame to be that of the last pose of the prior submap, and we only need to solve for the scale factor. We are happy to discuss this question further if this response has not fully addressed the question.
>
> [VGGT reconstruction with more than a projective ambiguity] As discussed in an earlier comment, the Projective Reconstruction Theorem from classical multi-view geometry says that if a set of uncalibrated cameras satisfies multi-view constraints, then the difference between the estimated reconstruction of the scene and the true reconstruction can be explained by a 15 DOF projective transformation. Points that do not satisfy multi-view constraints would be outlier points. To handle outliers, we filter out points using VGGT’s estimated confidence value for each point and use RANSAC in our estimation of relative homography estimates. We will also add to the limitations section that lens distortion is not handled by our projective rectification.

---

> > ### Comment · Reviewer_wLsM · 2025-08-05
> >
> > Dear Authors,
> >
> > Thank you for the response. Here are my remaining concerns regarding my initial questions:
> >
> > ### 1. **How frequently the VGGT reconstruction is distorted and how well the SL(4) homography transformation can model this distortion?**
> >
> > There are several ways to quantify these distortions. For example, we can measure how the point cloud itself is distorted by computing the difference between the ground-truth and the estimated point cloud after correcting the scale (using the median scale ratio of *estimated / ground-truth*).
> > To assess how well the SL(4) homography transformation can handle this distortion, we can estimate a residual SL(4) homography as a residual transformation and measure how much the residual is reduced. This process is essentially the same as the transformation estimation in this paper but applied between one VGGT estimation and the ground truth. The Projective Reconstruction Theorem surely holds if the point cloud at the overlapped frame is up to a homography transformation of the underlying ground truth, but we need to verify this quantitatively.
> >
> > ### 2. **Transformation composition and decomposition**
> >
> > Thank you for the response, but it has somewhat added to my confusion.
> >
> > * “Instead, we get the pose estimates from VGGT and apply a transforming homography to the poses.”
> >   This statement is confusing because in `solver.py:301`, the relative homography is **directly** estimated from two sets of point clouds, rather than as a residual relative homography on top of the SIM(3) poses from VGGT.
> >
> > * After `H_world_map` is estimated from `H_relative` via a pose graph, it is composed with the pose from VGGT (see `submap.py:63`); however, I do not see corresponding text for this line of code in the paper. In line 192, the homography is a relative one, not a global one. I assume the corresponding process should be elaborated at the end of Section 4.4.
> >
> > * Additionally, in `solver.py:282`, why is the SIM(3) transformation taken directly from the VGGT SE(3) with an additional averaged scale? For a fair comparison, it would be better to re-estimate the SIM(3) using the minimal solution derived from point clouds.
> >
> > ### 3. **Distinguishing between SIM(3) transformation and SIM(3) optimization**
> >
> > I would like to elaborate further on my original question to avoid confusion. Since the final product of the pipeline is always an SE(3) pose trajectory in a global coordinate frame, the inadequacy of SIM(3) may stem from the estimation method itself—meaning we may not reach the optimal SIM(3) that could lead to better performance. Therefore, it is worth discussing alternative approaches to obtain a better SIM(3). For example, on top of the SIM(3) from VGGT we can non-linearly refine the pose on the inlier point clouds, or—more closely related to this paper—we can initialize SIM(3) using a decomposed homography (at either the relative or global stage) and then refine it.

---

> > > ### Author Response · Authors · 2025-08-06
> > >
> > > 1. We would like to ask clarification about the reviewer’s concern for the use of the Projective Reconstruction Theorem. The theorem itself does not require quantitative proof since it is well proven from classical computer vision. Given its assumptions are met, it shows that any scene reconstructed by uncalibrated cameras differs from the true geometry by at most a projective transformation. If no extra information is provided, the best that can be done is a projective transformation. Points (except those that correspond to the epipoles) that violate the theorem's assumptions are outlier points and a projective transformation cannot reconcile them. The reviewer’s proposed experiment with measuring the residual between point clouds corrected by a homography would only show how well VGGT avoids producing outlier points (such as noisy points or hallucinating part of a scene). There are certainly examples of scenes where VGGT fails, and for these a homography or any practical transformation would not reconcile the scene. There are other scenes where the reconstruction is valid (in the sense that multi-view constraints from the fundamental matrix are satisfied) and these can corrected by a homography, or if VGGT is able to estimate a metric reconstruction (differing from the true scene by at most a similarity transform), then a Sim(3) transformation is sufficient. The examples in Figures 1,4,5, and 6 provide visual examples of when SL(4) is able to reconcile a scene when Sim(3) is not sufficient, which is used to demonstrate examples of projective ambiguities.
> > >
> > > 2a. The nodes in the factor graph in VGGT-SLAM correspond to the first keyframe of each submap, not all the keyframes in all submaps. H_relative (which is between the factor graph nodes) is thus between submaps. When we say that the poses come from VGGT, we mean that the poses for the keyframes inside a submap are from VGGT and we apply the corrective homography to each pose inside the respective submap. This is what corresponds to the subsection “Transformation of camera poses via homography” in 4.2 when we show how to apply the homography to a camera matrix $P$, where $P$ “is the camera matrix created from the poses and intrinsic estimates from VGGT.”
> > >
> > > 2b. This is related to our discussion of 2a as submap.py:63 is performing the computation in the section “Transformation of camera poses via homography.” The calculation self.vggt_intrinscs @ np.linalg.inv(self.poses)[:,0:3,:] computes the camera matrices $P$ (where the number of these is the number of keyframes in the submap) and these are then transformed to the global map frame using the homography self.H_world_map. The reviewer mentions that the homography $H_j^i$ in the manuscript line 192 is a relative homography, which is correct. Section 4.2 shows how to transform the poses with a homography, for which we show the homography $H_j^i$ used to align poses from frame j to frame i with $P_i = P_j {H_j^i}^{-1}$. The line submap.py:63 is aligning to the world frame to visualize all points in a global map in Viser, and uses the same equation where the homography is H_{world}^j
> > >
> > > 2c. The code in solver.py:282 follows lines 263-265 in the manuscript which, referring to Sim(3), says “we align relative rotation and translation between submaps using pose estimates from VGGT and estimate a scale correction by comparing the estimated point clouds of the overlapping frames.” The key point here is that the overlapping frames between submaps are the same image, hence at the same pose. Thus, in Sim(3) mode we can enforce this directly by setting the global translation and rotation of the first keyframe of submap j to be the global translation and rotation of the last keyframe in the prior submap (submap i). The only factor that must be solved for is the scale factor which comes from comparing the relative depth maps of the overlapping frames.
> > >
> > > 3. We appreciate the reviewer’s concern and here we discuss the measures which have been taken to address it. Our current Sim(3) code takes steps to filter outlier points in solver.py:291 since the point clouds are filtered by “[good_mask]”. This mask comes from the confidence threshold in VGGT. During development, we also experimented with other alternatives for Sim(3) such as using VGGT’s point cloud head vs its depth head, using median depth vs mean depth, and using a completely different robust estimation method, TEASER++ (which directly aligns the point clouds with a similarity transformation). We found that these showed insignificant performance differences. Since our paper’s main focus is SL(4), we did not include an ablation of the different ways to find the Sim(3) transform, although there are trivially multiple options such as the ones discussed here. We note that since the true poses of the overlapping frames are at the same pose (as mentioned in 2c), our current Sim(3) implementation has the benefit of enforcing this.

---

> ### Comment · Area_Chair_RVse · 2025-08-04
> **Dear Reviewer Please Engage in Discussion (2nd Reminder)**
>
> Dear Reviewer,
>
> The authors have replied to your concerns, and it would be very helpful if you could engage in a discussion to clarify your concerns.
>
> Best,
>
> AC

---

> ### Comment · Reviewer_wLsM · 2025-08-08
>
> Dear Authors:
> Here are some clarification of my questions:
> - 1.**Projective Reconstruction Theorem**: I am not asking verification of the theorem (as my in comment "The Projective Reconstruction Theorem surely holds") itself but asking about how well the the 4x4 homography can model the transformation between the two 3D point clouds on the same image but from two local maps of VGGT.  The reason to ask the question is because 4x4 homography has more DoF than SIM(3) transformation. If the underlying distortion is more than a 4x4 homography can handle (there is no guarantee) then a less DoF transformation is more regularized and potentially better. Given that solver.py:282 the SIM(3) transformation is directly from VGGT prediction instead of re-estimating from point clouds as the homography, we need to be more careful about the conclusion of transformations.
> - 2.c: Following the above question, it is better not to assume that the output from VGGT is the already the best SIM(3) (if there is one) transformation between two point clouds, this is why a re-estimation is needed.
> - 2.a&b: I think the confusion here is mainly brought by the index in line 191~193. The current index only present different local maps, to reuse the same notation to transform from local maps to the global map, it is better to make the index more general.

---

> > ### Author Response · Authors · 2025-08-08
> >
> > We thank the reviewer for providing clarification and answer each of the remaining questions as follows:
> >
> > We thank the reviewer for providing clarification on this question. We emphasize that the VGGT reconstruction should not differ more than a projective ambiguity except for outliers points, parts of the scene that have been hallucinated, or if the camera has lens distortion. In the case of outliers (which are present in most all scenes), we show that even for scenes where a similarity transform is sufficient, such as examples from 7-Scenes in Table 1, that our 5-point RANSAC with SL(4) performs either comparable or slightly better despite the extra degrees of freedom. In the case of the scene being hallucinated, this would be a failed reconstruction from VGGT which in our current work is out of scope to handle. It’s very possible that the extra degrees of freedom of SL(4) could offer some additional correction ability. Even though we think this is out of scope, to directly address the reviewer’s question we tried finding scenes where VGGT’s reconstruction completely fails, but we have not been able to find an example with only two submaps where Sim(3) gives a better alignment than SL(4).  For the third case of lens distortion, VGGT assumes minimal distortion in the model inputs, so passing in images from say a fisheye camera would not work in VGGT and hence also not work in VGGT-SLAM, which is why we assume rectified images similar to MASt3R-SLAM.
> >
> > 2c. During development, we tested multiple ways to perform Sim(3) alignment and the one which is most relevant to the reviewer’s suggestions is our use of TEASER++ to directly estimate a Sim(3) alignment between corresponding 3D points from each submap. As discussed in our prior response to question 3, we did not observe a significant difference with our current Sim(3) implementation and thus used our simpler Sim(3) baseline for the manuscript. The current implementation also globally optimizes over the relative Sim(3) estimates when loop closures are present, similar to the SL(4) variant, for fair comparison.
> >
> > 2a and 2b. We thank the reviewer for this suggestion and to fully address this point we will add clarification to our paper in this section to make the indices more general.

---

> > > ### Comment · Area_Chair_RVse · 2025-08-08
> > >
> > > Dear Reviewer wLsM,
> > >
> > > Today is the last day of discussions with the authors. The authors have provided a response to your recent concerns. Please engage at the earliest convenience and determine if it clarified your concerns.
> > >
> > > Best,
> > >
> > > AC

---

> ### Comment · Reviewer_wLsM · 2025-08-08
>
> Dear Authors:
> Thank you for the reply and it  has addressed my concerns and I will raise my final rating.
> - **SL(4) v.s. SIM(3)** : Thank you for dig deeper to find cases where SL(4) is insufficient while still outperform SIM(3), would love to see some examples in the appendix of the final version.
> - **How the SIM(3) transformation is estimated.** Will appreciate if a explicit statement can be added to reduce the potential question from the audience.

---

> > ### Author Response · Authors · 2025-08-08
> >
> > We thank the reviewer for their detailed feedback and discussion, and we will make the recommended additions in our revised final version.

---

### Official Review · Reviewer_enTd · 2025-07-01

**Clarity:** 3
**Significance:** 2
**Originality:** 3
**Rating:** 4
**Confidence:** 4

**Summary:**

This paper presents VGGT-SLAM, the first dense RGB SLAM system based on SL(4) manifold optimization. To address the projection ambiguity problem of uncalibrated monocular cameras, the authors use VGGT feedforward reconstruction to generate subgraphs, achieving global alignment by estimating a 15-degree-of-freedom homography matrix and combining it with factor graph optimization. Experiments show that this method outperforms traditional Sim(3) optimization in long sequence scenarios, particularly in cases of significant projection ambiguity, although it exhibits degradation issues in planar scenes.

**Questions:**

1.The 15 degrees of freedom in SL(4) optimization increase the difficulty of solving. In SL(4) optimization, how do you balance computational efficiency with accuracy? Have you considered a dynamic switching strategy between Sim(3) and SL(4) optimization?
2.If there are a large number of outliers in the depth map generated by VGGT (such as in low-texture areas), this situation is possible. Even with the RANSAC algorithm, the estimation of the homography matrix may still be erroneous.  Have you conducted research addressing this specific issue?
3.The paper mentions GPU limitations for VGGT (~60 frames). How does VGGT-SLAM perform in terms of memory usage and running speed when processing long sequences?

**Ethical Concerns:**

["NO or VERY MINOR ethics concerns only"]

**Final Justification:**

The authors' rebuttals adequately addressed my concerns. I suggest borderline accept of the paper.

**Limitations:**

Yes

**Paper Formatting Concerns:**

Not found

**Quality:**

3

**Strengths And Weaknesses:**

Strengths:
1.Introduces SL(4) manifold optimization to SLAM for the first time, addressing projection ambiguity in uncalibrated cameras and filling the gap in feedforward reconstruction methods for large-scale scenes.
2.Combines classic projection reconstruction theorems to prove the necessity of SL(4) transformations, achieving global consistency through factor graph optimization.
3.Validates the advantages of SL(4) over Sim(3) on the 7-Scenes and TUM datasets, with ablation studies analyzing the effects of loop closure and confidence thresholds.
Weaknesses:
1.Limited performance gain on 7-scenes dataset. Compared to using Sim(3)-based optimization, SL(4)-based optimization does not achieve noticeable improvement according to Table 1 and 3. The paper mentions that the homography estimation can easily degrade in planar scenes like TUM floors, leading to reconstruction divergence. However, the performance gain on TUM-RGBD dataset is higher. The analysis appears to be inconsistent.
2.Limited experimental evaluation. While paper emphasizes dense mapping, it is necessary to evaluate 3D mesh reconstruction quality quantitively and qualitatively. Importantly, the paper lacks computational cost comparison, such as tracking and mapping speed.

---

> ### Author Rebuttal · Authors · 2025-07-31
>
> We thank the reviewer for positive comments such as the significance of using SL(4) optimization to address projective ambiguity. We address each of the reviewer’s questions and concerns below:
>
> [SL(4) performance on benchmarks] We emphasize that while our proposed simpler Sim(3)  baseline approach is sometimes able to perform well, showing good performance on 7-Scenes and TUM RGB-D overall, there are multiple important cases where Sim(3) is insufficient to align VGGT submaps due to a projective ambiguity which we reconcile with our full SL(4) method. To highlight this, we include visual examples on scenes where SL(4) is needed, such as in Figure 1, and include three additional examples in the Supplementary Material. On the nine TUM RGB-D scenes, the scene where the SL(4) version of VGGT-SLAM has the largest error is on the floor scene which from analyzing the optimization steps occurs due to planar degeneracies when estimating the 15 DOF homography, which is why we mention that scene as an example limitation of planar degeneracies. This is consistent with the well-known issue that estimating the full 15 DOF homography matrix is degenerate in the case of planar points (such as in Multi-View Geometry [25], chapter 10) since 5 unique points are required for a unique solution.
>
> To further show the benefits of SL(4), in our revised paper we will also include an additional plot showing the fluctuations in VGGT’s estimate of the camera intrinsics. These fluctuations can be attributed to the projective ambiguity motivating the need for a projective alignment. We summarize some of the results of VGGT’s estimate of the focal length (fx in pixels) of all keyframes in a scene, where a single camera is used for all images in a scene and thus should have constant focal length. 7-Scenes - avg: 435.1, std: 9.0, range: 59.7; Tabletop (Figure 5 in Supplementary Material) - avg: 679.4, std: 37.7, range: 117.3; Outdoor (Figure 6 in Supplementary Material) - avg: 738.9, std: 51.8, range: 177.3; Office Loop (Figure 2) - avg: 429.4, std: 6.9, range: 33.4. In particular, for the Tabletop and Outdoor scenes (for which in our paper we qualitatively demonstrate improvement with SL(4) over Sim(3)), there is substantial discrepancy in the focal length estimates from VGGT.
>
> [Computational Cost evaluation] We thank the reviewer for mentioning this important evaluation that should be included. We have conducted runtime analysis for VGGT-SLAM and will add the timing results to the paper. On a 4090, VGGT-SLAM with SL(4) for submap size 16 takes 991.6 ms per submap with the following timing breakdown (all in milliseconds). VGGT: 820.2 ms, keyframe detection: 90.7, loop closure detection and SALAD computation: 41.8 ms, relative homography estimation: 31.4 ms, and backend optimization: 7.4 ms. For Sim(3), all steps are the same except the relative estimation takes 11.3 ms instead of 31.4 ms. For mesh reconstruction, our method produces point clouds which we evaluate following the dense evaluation protocol of MASt3R-SLAM in Table 3. We also provide qualitative results in Figures 1 and 2 and in Figures 4–11 in the Supplementary Material.
>
> [Balancing SL(4) computation and accuracy] We thank the reviewer for mentioning this important detail about balancing computation between SL(4) and Sim(3). Both Sim(3) and SL(4) use a nonlinear backend optimization (using GTSAM), where the difference is on which manifold the optimization occurs. The relative estimation of SL(4), while more involved that Sim(3), is still a linear optimization problem and after conducting timing analysis, we observe the computation cost for SL(4) only uses about 20.1 ms more than Sim(3) which is relatively small compared to the time to run VGGT. The reviewer raises an interesting idea to dynamically switch between Sim(3) and SL(4) and this is something we have previously experimented with during development by comparing the relative inliers and outliers from each method and picking the optimized solution accordingly. This is an idea which is used in classical structure from motion, for example COLMAP uses it to switch between 3x3 homography and essential matrix estimation, and it is possible that better tuned inlier detection could lead to future improvements in this direction.
>
> [Handling outliers] To handle outlier points, particularly in challenging low texture regions as mentioned by the reviewer, during development we did experiment with different RANSAC parameters (such as outlier thresholds and number of iterations) and with confidence thresholds to filter points from VGGT, the latter of which helps to increase the inlier ratio. We keep the selected parameters constant across all quantitative experiments to show our default parameters are sufficient to handle outliers in most cases. A 5-point RANSAC is known to statistically require only 145 iterations (although we use 300 max iterations) for a 99% success probability with 0.5 outlier ratio and scales well for higher outlier ratios. VGGT has high resilience in low texture environments compared to prior methods, but it is expected that in extreme cases with high outliers most SLAM systems will not be able to produce a valid solution. To improve RANSAC estimation, we also experimented with more complex point sampling such as weighted sampling based on the VGGT confidence estimate and weighted sampling based on the level of non-coplanarity between points. However, we empirically found these approaches did not bring significant improvement to justify the added complexity.
>
> [Performance on long sequences] We thank the reviewer for raising this important question. Since VGGT-SLAM works by aligning submaps, computation time is roughly constant as scene size increases. From the timing breakdown of VGGT-SLAM’s components discussed earlier, SALAD feature extraction and VGGT are run over constant number of frames (the size of the submap), the relative estimation of SL(4) is between two submaps which is constant, and the backend is a small factor graph (typically, real time mapping systems with GTSAM have on the order of 10s of thousands of factors for bearing measurements while we just have factors between adjacent submaps and submaps with loop closures). Memory of the core part of VGGT-SLAM with the exception of SALAD retrieval vectors is also approximately constant.  In practice, we do store the point clouds of each submap in working memory for visualizing map construction in real-time and this does increase with map size. Since the visualization is not essential for VGGT-SLAM and can be disabled, the point clouds could either be stored in system memory or engineered to be more memory efficient depending on the application.

---

> > ### Comment · Reviewer_enTd · 2025-08-04
> > **Official Comment by Reviewer enTd**
> >
> > My questions are adequately addressed by the authors' response.

---

> > ### Comment · Reviewer_enTd · 2025-08-07
> > **Official Comment by Reviewer enTd**
> >
> > Thank you for the thorough rebuttal, and I am maintaining my positive rating.

---

### Decision · Program_Chairs · 2025-09-17

**Decision:**

Accept (poster)

**Comment:**

This paper introduces VGGT-SLAM, a dense RGB SLAM system that incrementally and globally aligns submaps using uncalibrated monocular cameras. Unlike prior methods relying on similarity transforms, it leverages SL(4) manifold optimization to estimate 15-DOF homography transformations, addressing reconstruction ambiguity inherent to uncalibrated setups.

The initial concerns of the reviewers can be summarized as follows:

1. Limited experiments (e.g., computational analysis, inaccurate intrinsics) and inconsistent conclusions from experiments (R-enTd, R-wLsM, R-Wbeo).
2. Questionable approach stating that SIM(3) are insufficient (R-wLsM); and sensitivity to outliers and degenerate cases (R-BmpF).

After the discussion between authors and reviewers, most reviewers felt that the rebuttal and discussion clarified most of their concerns. Because the paper presents interesting contributions, and reviewers are supportive of accepting this work, I am recommending acceptance. We encourage the authors to include clarifications from the discussion with the reviewers to a final version.